# CoMPS: Continual Meta Policy Search

## Abstract

We develop a new continual meta-learning method to address challenges in sequential multi-task learning. In this setting, the agent's goal is to achieve high reward over any sequence of tasks quickly. Prior meta-reinforcement learning algorithms have demonstrated promising results in accelerating the acquisition of new tasks. However, they require access to all tasks during training. Beyond simply transferring past experience to new tasks, our goal is to devise continual reinforcement learning algorithms that learn to learn, using their experience on previous tasks to learn new tasks more quickly. We introduce a new method, continual meta-policy search (CoMPS), that removes this limitation by meta-training in an incremental fashion, over each task in a sequence, without revisiting prior tasks. CoMPS continuously repeats two subroutines: learning a new task using RL and using the experience from RL to perform completely offline meta-learning to prepare for subsequent task learning. We find that CoMPS outperforms prior continual learning and off-policy meta-reinforcement methods on several sequences of challenging continuous control tasks.

## 1 Introduction

Meta-reinforcement learning algorithms aim to address the sample complexity challenge of conventional reinforcement learning (RL) methods by *learning to learn* – utilizing the experience of solving prior tasks in order to solve new tasks more quickly. Such methods can be exceptionally powerful, learning to solve tasks that are structurally similar to the meta-training tasks with just a few dozen trials (Finn et al., 2017; Duan et al., 2016; Wang et al., 2016; Zintgraf et al., 2020). However, prior work on meta-reinforcement learning is generally concerned with asymptotic meta-learning performance, or how well the meta-trained policy can adapt to a single new task at the end of a long meta-training period. The meta-training process itself requires iteratively attempting each meta-training task in a "round-robin" fashion. While this is reasonable in supervised settings, in reinforcement learning revisiting and repeatedly interacting with previously seen tasks in the real world may be costly or even impossible. For example, when learning to tidy in different homes – effective generalization requires visiting many homes and interacting with many items, but needing to revisit every prior home and item on each iteration of meta-training would be impractical. Instead, we would want the robot to use each new experience in each new home to *incrementally* augment its skillset so that it can acquire new cleaning skills in new homes more quickly, as shown in Figure 1 using examples from MetaWorld (Yu et al., 2020b). In this paper, we study the continual meta-reinforcement learning setting, where tasks are experienced one at a time, without the option to collect additional data on previous tasks. The objective is to decrease the time it takes to learn each successive task, as well as achieve high asymptotic performance.

This paper proposes a novel meta-learning algorithm for tackling the continual multi-task learning problem in a reinforcement learning setting. The key desiderata for such a method are the following. First, an effective continual meta-learning algorithm should adapt quickly to new tasks that resemble previously seen tasks, and at the same time adapt (even if slowly) to completely novel tasks. This adaptation is crucial in the early stages of meta-training when the number of tasks seen so far is small, and every new task appears new and different. Second, an effective continual meta-learning algorithm should be able to use all previously seen tasks to improve its ability to adapt to future tasks, integrating information even from tasks seen much earlier in training, and are therefore far off-policy. To address

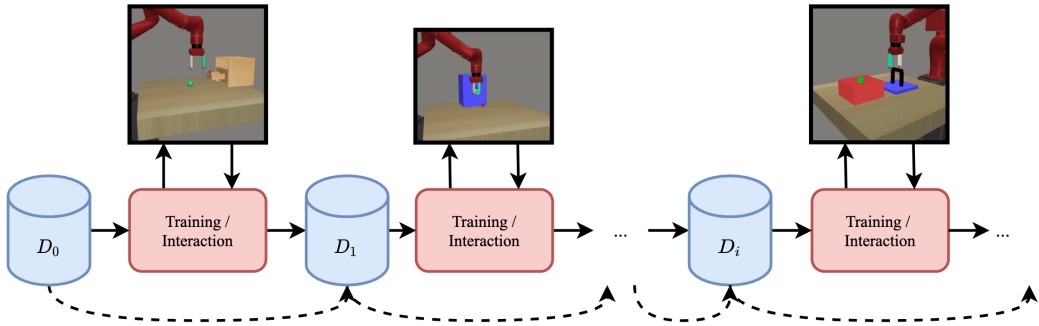

Figure 1: In the continual meta-RL setting, the agent interacts with a single task at a time and, once finished with a task, never interacts with it again. To complete the full sequence of tasks efficiently, the agent must reuse experience from preceding tasks to more quickly adapt to subsequent tasks.

the first requirement, we base our approach on model-agnostic meta-learning (MAML) (Finn et al., 2017). MAML adapts to new tasks via gradient descent, while the meta-training process optimizes the initialization for this gradient descent process to enable the fastest possible adaptation. Because the adaptation process in MAML corresponds to a well-defined learning algorithm, even new out-of-distribution tasks are learned (albeit slowly), while tasks that resemble those seen previously will be learned much more quickly (Finn & Levine, 2018). To address the second requirement, and make it possible to incorporate data from older tasks without revisiting them, we devise an off-policy method where the adaptation process corresponds to on-policy policy gradient. This meta-training uses behavioral cloning on successful episodes experienced by the agent from older tasks. Although the "inner loop" policy gradient adaptation process is on-policy, and the agent adapts to each new task with on-policy experience, the meta-training process, which is similar to distillation of previously collected experience, is off-policy. Essentially, the agent meta-trains the model such that a few steps of policy gradient result in a policy that mimics the most successful episodes on each previously seen task. This can effectively enable our approach to incorporate experience from much older policies and tasks into the meta-training process. Although our algorithm is intended to operate in settings where new tasks are revealed sequentially, one at a time, as in the home cleaning robot example before, it still relies on storing experience from prior tasks, similar to rehearsal-based methods (Isele & Cosgun, 2018; Riemer et al., 2018; Rolnick et al., 2019; Atkinson et al., 2021), but instead uses this information to accelerate sequential task acquisition.

Our primary contribution is a meta-reinforcement learning algorithm that supports the sequential multi-task learning setting, where the agent cannot revisit previous tasks to collect data. To evaluate our approach, we modify a collection of commonly used meta-RL benchmarks into continual multi-task problems, with tasks presented one at a time. Our method outperforms other methods, achieving a higher average reward with fewer samples on average over each of the tasks in the sequence. In addition, we evaluate each method's ability to generalize over a collection of held-out tasks during training. We find that CoMPS achieves a higher meta-test time performance on held-out tasks. Our experiments show that as the agent experiences more tasks, learning time on new tasks decreases, indicating that meta-reinforcement learning performance increases asymptotically with the number of tasks. To our knowledge, our work is the first to formulate and address the continual meta-reinforcement learning problem, which we propose is a realistic formulation of meta-RL for real-world settings, where prior tasks cannot be revisited after they have been solved. While several prior methods can be used in this setting, we show in our experiments that they perform significantly worse than CoMPS.

## 2 RELATED WORK

Meta-learning, or *learning-to-learn* (Schmidhuber, 1987; Bengio et al., 1990; Thrun & Pratt, 2012), is concerned with the problem of learning a prior, given a set of tasks, that enables more efficient learning in the future. We focus on meta-learning for reinforcement learning (Schmidhuber, 1987; Duan et al., 2016; Wang et al., 2016; Finn et al., 2017; Mishra et al., 2017). There are many ways to represent the meta-learned model, including black box models (Duan et al., 2016; Wang et al., 2016; Mishra et al., 2017; Stadie et al., 2018; Rakelly et al., 2019; Zintgraf et al., 2020; Sæmundsson et al.,

2018; Fakoor et al., 2019; Sun et al., 2019; Dorfman & Tamar, 2020), applying gradient descent from initial parameters (Finn et al., 2017; Gupta et al., 2018b; Rothfuss et al., 2018; Clavera et al., 2018; Mendonca et al., 2019; Zintgraf et al., 2019; Mitchell et al., 2020), and training a critic to provide policy gradients (Sung et al., 2017; Houthooft et al., 2018; Yu et al., 2018; Chebotar et al., 2019). Recent work has made progress toward using supervised meta-learning in an online setting (Ren et al., 2018; Finn et al., 2019; Khodak et al., 2019; Jerfel et al., 2019; Grant et al., 2019; Zhuang et al., 2019; Antoniou et al., 2020; Wallingford et al., 2020; Yu et al., 2020a; Yao et al., 2020). Adaptation without task boundaries or inside of an RL episode has also become a new area of investigation for meta-learning (Nagabandi et al., 2019; Javed & White, 2019; Harrison et al., 2019; Al-Shedivat et al., 2018; He et al., 2019; Harrison et al., 2019). Our work focuses on a distinct problem setting, where for meta-training the RL tasks are experienced sequentially, and the goal is to learn each RL task more quickly by leveraging the experience from the prior tasks, without the need to revisit prior tasks. The general on-policy meta-RL setting with infinite task sets also corresponds to the continual learning problem as well, however, the finite task setting is more representative of the sorts of situations that RL agents are likely to encounter in the real world that benefit from generalization.

Continual or online learning studies the streaming data setting, where experience is used for training as soon as it is received (Thrun & Pratt, 2012; Hazan et al., 2016; Chen & Liu, 2016; Parisi et al., 2019; Khetarpal et al., 2020). Both terms describe the process of learning tasks in sequence while avoiding the problem of *forgetting* (French, 1999; Rusu et al., 2015; Parisotto et al., 2016; Rusu et al., 2016; Chen et al., 2015; Tessler et al., 2016; Aljundi et al., 2019). Recent meta-learning based methods have made inroads on the continual meta-learning but they are limited to non-RL problems (K J & N Balasubramanian, 2020; Gupta et al., 2020). Following Rolnick et al. (2019), we do not explicitly aim to address the problem of forgetting, and instead retain data from prior tasks in a replay buffer to use for meta-RL training a model that can adapt quickly to new tasks. Other recent methods support continual learning without ground truth task boundaries (Schmidhuber, 2011; Srivastava et al., 2012; Nguyen et al., 2017), but have not yet been demonstrated to perform well in reinforcement learning settings. Even with replay buffers, the data from previous tasks is still challenging to reuse since it was collected by a different policy.

Off-policy RL methods are known to achieve good sample efficiency by reusing prior data. Therefore, several recently proposed sample-efficient meta-RL algorithms have been formulated as off-policy methods (Rakelly et al., 2019; Fakoor et al., 2019; Nagabandi et al., 2019; Sæmundsson et al., 2018). In principle, these methods could be extended to the continual meta-learning setting. However, in practice, their ability to utilize data from past tasks collected under an older policy is limited, and we find, via our experiments, that they tend to perform poorly in the continual meta-reinforcement learning setting, possibly due to their limited ability to extrapolate to the new out of distribution tasks.

To meta-train without revisiting prior tasks, our method uses a type of self-imitation. Although this resembles imitation learning, it does not use external demonstrations: meta-self-imitation learning uses high reward experience the agent itself collected in prior tasks. There are a number of non-meta-learning based methods that used behavioral cloning, distillation, and self-imitation for re-integrating previous experience (Rusu et al., 2015; Parisotto et al., 2016; Levine et al., 2016; Teh et al., 2017; Ghosh et al., 2018; Berseth et al., 2018). Our work uses meta-self-imitation learning as the outer objective. As such not only trains a model to imitate a previous policy but also trains this policy to adapt given little data quickly. The actual meta-learning procedure in CoMPS resembles GMPS, but with several important changes that empirically lead to significant improvement in the continual meta-learning setting: (1) CoMPS uses a PPO-based inner policy gradient formulation for meta-rl training and RL-based task learning, where GMPS uses vanilla policy gradient for meta-rl training and off-policy SAC (Haarnoja et al., 2018) to train expert policies; by using SAC, a non-policy gradient-based RL algorithm, GMPS can not train expert policies on new tasks using the meta-rl-trained policy parameters to accelerate learning. (2) CoMPS meta-trains using only off-policy data. While these components use prior techniques (Espeholt et al., 2018), their integration in CoMPS and their application to continual meta-reinforcement learning is novel.

## 3 PRELIMINARIES

**Reinforcement learning.** RL problems are generally formalized as a *Markov decision process* (MDP), defined by the tuple MDP $= (\mathcal{S}, \mathcal{A}, \mathcal{P}, R, \rho, \gamma, T)$, with the state space $s \in \mathcal{S}$, the action

space $a \in \mathcal{A}$, a transition probability function $\mathcal{P}(s'|s, a)$, a reward function $R(s, a)$, an initial state distribution $\rho(s_0)$, a discount factor $\gamma \in (0, 1]$, and a time horizon $T$. The agent's actions are defined by a policy $\pi(a|s, \theta)$ parametrized by $\theta$. The objective of the agent is to learn an optimal policy: $\theta^* := \operatorname{argmax}_\theta J(\theta)$, where $J(\theta) = \mathbb{E}_{s_{t+1} \sim \mathcal{P}(\cdot|s_t, a_t), a_t \sim \pi(\cdot|s_t; \theta), s_0 \sim \rho}[\gamma^t R(s_t, a_t)]$ is the expected discounted return.

**Meta-reinforcement learning.** Meta-learning aims to leverage a set of meta-training tasks to enable fast adaptation on a different set of meta-test tasks not seen during training. MAML (Finn et al., 2017) accomplishes this by meta-training a set of initial parameters $\theta$ over the training tasks to efficiently adapt to a new task. In meta-reinforcement learning, each RL task $\mathcal{T}_i$ is a different MDP, with its own task objective $J_i$, defined as before. The state $\mathcal{S}$ and action space $\mathcal{A}$ are the same for these MDPs, however their transitions, rewards, and initial states can differ. This meta-training process itself is sample inefficient (even though meta-test time adaptation is fast), requiring on-policy trajectories to estimate the inner policy gradient and many more trajectories for the outer objective. To reduce the cost of needing additional trajectories for the outer objective, Mendonca et al. (2019) propose meta-training with the expected discounted return as the inner task loss and supervised imitation as the outer loss:

$$\min_\theta \sum_{\mathcal{T}_i} \sum_{\mathcal{T}_i^v \sim \mathcal{D}_{0:i}^*} \mathbb{E}_{\mathcal{T}_i}[\mathcal{L}_{BC}(\theta + \alpha \nabla_\theta J_i(\theta), \mathcal{D}_i^v)], \quad \mathcal{L}_{BC}(\theta_i, \mathcal{D}_i) = -\sum_{(s_t, a_t) \in \mathcal{D}_i} \log \pi(a_t|s_t, \theta_i). \quad (1)$$

The outer behavioral cloning loss $\mathcal{L}_{BC}$ does not require collecting more data from the environment, but on-policy data from the environment is needed for computing the inner update on the policy parameters $\phi_i = \theta + \alpha \nabla_\theta J_i(\theta)$. This meta-RL method is more sample efficient when we have near-optimal data for the outer behavioral cloning loss $\mathcal{L}_{BC}$, but it cannot be trivially extended to a continual setting, the inner objective requires data to be repeatedly collected from each meta-training task. The following section will outline the continual multi-task learning problem and describe how we can extend GMPS to such a setting, removing the need to revisit prior tasks and carefully include a process for the agent to generate its own near-optimal data.

## 4 CONTINUAL META POLICY SEARCH

In the continual multi-task reinforcement learning setting, which we study in this paper, an agent proceeds through many tasks $\mathcal{T}_i$, one at a time. The agent's goal is to quickly solve each new task using a learning rule $L$, achieving high reward as efficiently as possible. In order to accomplish this, the agent can use experience from solving tasks $\mathcal{T}_{0:i}$ to learn how to adapt quickly to each new task $\mathcal{T}_{i+1}$, but cannot revisit past tasks to collect additional data once it moves on to the next task. This differs from the standard reinforcement learning setup, in which there is a single task $\mathcal{T}_0$, and the standard meta-RL setting, where all tasks can

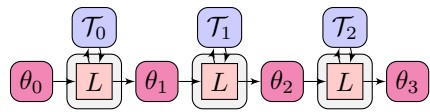

Figure 2: The continual multi-task reinforcement learning problem for a sequence of three tasks. The agent applies learning algorithm $L$ on task $\mathcal{T}_i$ and forwards policy parameters $\theta_{i+1}$ after each task.

be revisited as many times as needed during meta-training but is closer to and RL approximation of online meta learning (Finn et al., 2019). After each round of training on a new task, the agent produces a new set of policy parameters $\theta_i$ to serve as initialization for the next task. The learning rule $L$ needs to serve two purposes: (1) solve the current task; (2) prepare the model parameters for efficiently solving future tasks. This process is depicted in Figure 2.

**CoMPS overview.** CoMPS addresses the continual multi-task RL problem one task at a time through a sequence of tasks $\mathcal{T}_i$. The $L$ process for CoMPS consists of two main parts, a reinforcement learning $RL$ process to learn the new task involving potentially hundreds of training steps and a meta-RL $M$ process, which uses the off-policy experience from previous tasks to meta-train the initial parameters for $RL$. We illustrate the flow of these processes for CoMPS in Figure 3. The combination of $RL$ and $M$ creates a solution to the continuous multi-task RL problem that can perform non-trivial learning via meta-learning across tasks to accelerate learning. The $RL$ step consists of using an on-policy RL algorithm to optimize a policy on $\mathcal{T}_i$ (described next), which benefits from the previous rounds of meta-training and is therefore very fast. This RL training process produces a dataset of trajectories $\mathcal{D}_i = \{\tau_0, \dots, \tau_j\}$ where $\tau = \{(s_0, a_0, r_0), \dots, (s_T, a_T, r_T)\}$. From this dataset, we set aside the experience that achieved the highest reward on the task as $\mathcal{D}_i^* \leftarrow \max_{\tau \in \mathcal{D}_i} \sum_{(a_t, s_t) \in \tau} R_i(s_t, a_t)$ called the skilled experience.

The details of the $RL$ step, the algorithm used, and the implementation is given in Section 4.1. The meta-training in $M$ uses the experience collected during $RL$ in two separate data sets. The first dataset corresponds to the skilled experience $\mathcal{D}^*_{0:i}$, which is used in the outer meta-imitation learning objective. The second dataset consists of all the experience seen so far, $\mathcal{D}_{0:i}$, which is used for estimating the inner policy gradient based on Equation 1. Thus, instead of naïvely finetuning parameters on each new task, as in the case of standard continual RL methods, the $M$ step in CoMPS produces meta-trained parameters that are optimized such that all prior tasks can be learned as quickly as possible starting from these parameters. When there are enough such tasks, these meta-trained parameters can provide forward transfer and generalize to enable fast adaptation to new tasks, enabling them to be acquired more efficiently than with naïve finetuning. However, to accomplish this, the $M$ step must train from off-policy data from prior tasks. In Section 4.2, we describe how we implement this procedure.

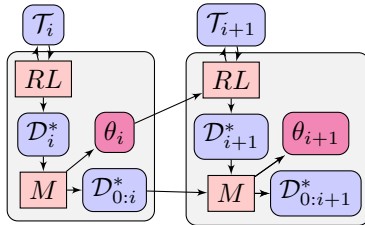

Figure 3: CoMPS is split in to two process: a reinforcement learning step $RL$ and a meta-learning step $M$. The meta-learning step $M$ uses the data gathered thus far, denoted $\mathcal{D}^*_{0:i}$, to meta-train $\theta_i$. $RL$ is initialized from these parameters $\theta_i$ for the next task.

## 4.1 TASK ADAPTATION VIA POLICY GRADIENT

To solve each task in the $RL$ step, we use the popular policy gradient algorithm PPO (Schulman et al., 2017). PPO uses stochastic policy gradients to determine how to update the policy parameters $\theta$ compared to a recent version of the parameters that generated the current data $\theta'$, using a distribution ration $r_t(\theta) = \frac{\pi(a|s,\theta)}{\pi(a|s,\theta')}$. A first-order constraint, in the form of a gradient clipping term, is used to limit on-policy distribution shift of $r_t$ while optimizing the policy parameters using $\mathcal{L}_{ppo}(\theta) = \mathbb{E}[\min(r_t(\theta)\hat{A}_{\pi_{\theta'}}, \text{clip}(r_t(\theta), 1 - \epsilon, 1 + \epsilon)\hat{A}_{\pi_{\theta'}})]$. The advantage $\hat{A}_{\pi_{\theta'}} = r_{t+1} + \gamma V_{\pi_{\theta'}}(s_{t+1}) - V_{\pi_{\theta'}}(s_t)$ is a score function that measures the improvement an action has over the expected policy performance $V_{\pi_{\theta'}}(s_t)$. CoMPS increases the sample efficiency of PPO in the $RL$ step via initialization from the network parameters $\theta_i$ from the $M$ step, which performs off-policy meta-RL, that we describe next. During the $RL$ step, for each episode we collect 20 trajectories and perform 16 training updates with a batch size of 256. Additional details, including the learning parameters and network design, can be found in Appendix C.

## 4.2 OUTER LOOP META-LEARNING

The $M$ process of CoMPS meta-trains a set of parameters $\theta_i$ using meta-self-imitation from the skilled experience. The outer self-imitation learning objective in Eq. 1 uses the skilled experience $\mathcal{D}^*_{0:i}$ to train the agent to be capable of (re)learning these skilled behaviors from one or a few policy gradient steps using previously logged off-policy experience, sampled randomly from $\mathcal{D}_{0:i}$. In contrast to methods that are concerned with forgetting, the parameters produced by this meta-RL training can quickly learn new behaviors that are similar to the high-value policies from previous tasks and, if enough prior

---

**Algorithm 1** CoMPS Meta-Learning

1: **require:** $\theta$, skilled $\mathcal{D}^*_{0:i}$ and off-policy $\mathcal{D}_{0:i}$
2: **for** $n \leftarrow 0 \dots N$ **do**
3:   **for** $j \leftarrow 0 \dots i$ **do**
4:     $\mathcal{D}^{tr}_j \leftarrow$ sample $m$ trajectories from $\mathcal{D}_j$
5:     $\phi_j \leftarrow \theta + \alpha \nabla J_j(\theta)$ (via imp. weights, subsection 4.3)
6:     Sample data $\mathcal{D}^{val}_j \sim \mathcal{D}^*_j$
7:     Update $\theta \leftarrow \theta - \beta \nabla \mathcal{L}_{BC}(\phi_j, \mathcal{D}^{val}_j)$
8:   **end for**
9: **end for**

---

tasks have been seen, likely generalize to quickly learn new tasks as well. The use of gradient-based meta-learning is particularly important here: as observed in prior work (Finn & Levine, 2018), gradient-based meta-learning methods are more effective at generalizing to new tasks under mild distributional shift as compared to contextual methods, making them well-suited for continual meta-learning with non-stationary task sequences, where new tasks can deviate from the distribution of tasks seen previously. However, in the continual setting, where prior tasks cannot be revisited, a significant challenge in this procedure is that the meta-RL optimization needs to estimate the policy gradient $\nabla_\theta J(\theta)$ in its inner loop for each previously seen task, without collecting additional data from the task (which it is not allowed to revisit). To address this challenge, we will utilize an

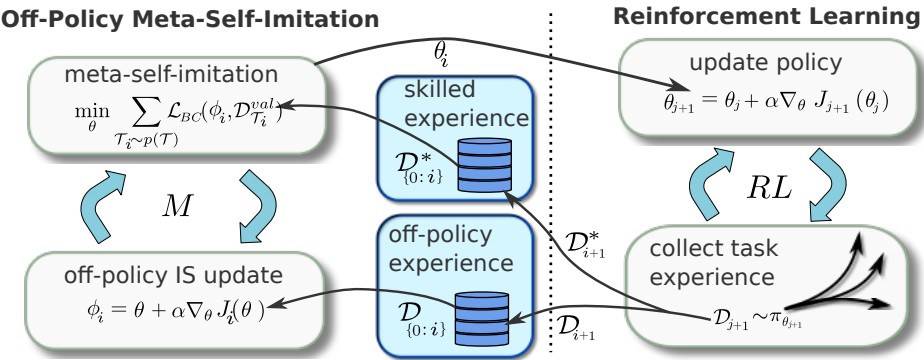

Figure 4: Outline of CoMPS. The left side corresponds to the $M$ block for Figure 3 and the right the $RL$ block. On the right ($RL$), for each round $i$ the policy $\pi_{\theta_i}$ is initalized using the previously trained meta-policy parameters. After $j$ iterations of RL training on task $i$, the experience for task $i$ is collected into the off-policy buffer $\mathcal{D}_{0:i}$ and skilled experience is stored in another buffer $\mathcal{D}^*_{0:i}$. This experience is given to $M$ that uses the off-policy experience for the inner expected reward updates. The outer step behavior clones from the skilled experience the $RL$ agent generated itself previously.

importance-sampled update that we describe in Section 4.3, using samples from the full dataset of off-policy experience for that task, $\mathcal{D}_{0:i}$, to estimate an inner loop policy gradient. In effect, this procedure trains the model to learn policies that are close to the near-optimal trajectories in $\mathcal{D}^*_{0:i}$ by taking (off-policy) policy gradient steps on the sub-optimal trajectories in $\mathcal{D}_{0:i}$. The complete meta-training process is summarized in Algorithm 1, and corresponds to a reinforcement learning inner update and a meta-imitation learning outer update, though imitation uses the agent's own experience without requiring any demonstrations. In our implementation, the skilled data for task $i$ consists of the 20 highest-scoring trajectories. We also found that only 5% of experience from prior tasks needed to be stored for good performance. Further details on the networks and hyperparameters used for Algorithm 1 are available in Appendix D.

### 4.3 Off-Policy Inner Gradient Estimation for Meta-Learning

In this section, we describe the particular form of the inner-loop policy gradient estimator used in Algorithm 1. Although many prior works have studied importance-sampled policy gradient updates, and GMPS (Mendonca et al., 2019) uses an importance-sampling update based on PPO (Schulman et al., 2017), we found this simple importance sampled approach to be insufficient to handle the highly off-policy data in $\mathcal{D}_{0:i}$. This is because the data for the earlier tasks may have been collected by substantially different policies than the data from the latest tasks. To enable our method to handle such highly off-policy data, we utilize *both* an importance-sampled policy gradient estimator and an importance-sampled value estimate for the baseline in the policy gradients. The former is estimated via clipped importance weights, while the latter uses an estimator based on V-trace (Espeholt et al., 2018) to compute value estimates, which are then used as the baseline. For state $s_m$ given a trajectory $(s_t, a_t, r_t)_{t=m}^{m+n}$, we define the $n$-step V-trace value targets for $V(s_m) = \sum_{t=0}^{n} \gamma^t r_t$, as:

$$v_m = V(s_m) + \sum_{t=m}^{m+n=1} \gamma^{t-m} \left( \prod_{i=m}^{t-1} c_i \right) \rho_t (r_t + \gamma V(s_{t+1}) - V(s_t)). \qquad (2)$$

The values $\rho_t = \min(\bar{\rho}, r_t(\theta))$ and $c_i = \min(\bar{c}, r_i(\theta))$ are truncated importance weights, where $\bar{\rho}$ and $\bar{c}$ are hyperparameters, and $r_t(\theta) = \pi(a_t|s_t,\theta)/\pi(a_t|s_t,\theta')$, where $\theta'$ denotes the parameter vector of the policies that sampled the trajectory in the dataset. The value function parameters $\omega$ are trained to minimize the $l2$ loss $|v_m - V_\omega(s_m)|$. The V-trace value estimate is then used to estimate the advantage values for policy gradient, which are given by $\hat{A}_m = r_m + v_{m+1} - V_\omega(s_m)$, and the gradient is then given by $\nabla_\theta J(\theta) \approx \frac{1}{N} \sum_i \rho_i \nabla_\theta \log \pi_\theta(a_i|s_i) \hat{A}_i$, analogously to PPO and other importance-sampled policy gradient algorithms. We use this estimator for the inner loop update in Algorithm 1 line 5. We show in our ablation experiments that this approach is needed to enable successful meta-training using the exhaustive off-policy experience collected by CoMPS.

## 4.4 CoMPS Summary

An outline of the entire CoMPS algorithm, including how data is accumulated as more tasks are solved, is shown in Figure 4. The $RL$ process keeps track of the set of trajectories that achieve the highest sum of rewards $\mathcal{D}_i^*$. The $RL$ step returns the separate skilled experience $\mathcal{D}_i^*$, and all experience collected during RL training in $\mathcal{D}_i$. The $M$ step uses this experience to perform meta-RL via training a model to learn how to reproduce the best policies achieved from previous tasks. Significantly, this meta-RL training process can accelerate the $RL$ process even in fully offline settings, allowing the agent to train a meta-RL model without collecting additional experience.

## 5 Experiments

Our experiments aim to analyze the performance of CoMPS on both stationary and non-stationary task sequences in the continual meta-learning setting, where each task is observed once and never revisited again during the learning process. To this end, we construct a number of sequential meta-learning problems out of previously proposed (non-sequential) meta-reinforcement learning benchmarks. We separate our evaluation into experiments with stationary task distributions, where each task in the sequence is sampled identically, and non-stationary task distributions, where the tasks either become harder over time or else are selected to be maximally dissimilar for prior tasks (see discussion below). We describe the task domains and the methods in our comparisons below, with further details provided in Appendix A.

**Tasks.** An illustration of the tasks in our evaluation is provided in Figure 5, along with a visualization of the tasks, and includes the following task families. In the **Ant Goal** environment, a quadrupedal robot must reach different goal locations, arranged in a semicircle in front of the robot. The non-stationary distribution selects the locations that are furthest from the previously chosen location each time, while the stationary one selects them at random. The **Ant Direction** tasks use the same quadrupedal robot that must now run in a particular direction. The non-stationary and stationary distributions are constructed as above. In **Half Cheetah** the goal is to control the half-cheetah to run at different velocities, either forward or backward. The direction is chosen randomly, but in the non-stationary distribution, the desired velocity magnitude increases over time. Last, we utilize the suite of robotic manipulation tasks from **MetaWorld** Yu et al. (2020b), which we arrange into a sequence. The non-stationary sequence orders the tasks in increasing difficulty, as measured by how long regular PPO can solve the tasks individually.

**Prior methods.** We compare CoMPS both to prior meta-learning methods and to prior methods for continual learning. Since learning without revisiting prior tasks requires an off-policy algorithm, we include PEARL (Rakelly et al., 2019) as a meta-learning baseline, which utilizes the off-policy SAC (Haarnoja et al., 2018) algorithm and can in principle learn without revisiting prior tasks. While several prior methods use policy gradients with meta-RL (Rothfuss et al., 2018; Al-Shedivat et al., 2018), these methods require on-policy data, making them unsuited for this continual meta-learning problem setting. For continual learning, we include a PPO transfer learning baseline (denoted PPO+TL), which trains sequentially on the tasks, as well as the more sophisticated Progress & Compress (PNC) method (Schwarz et al., 2018), which further guards against forgetting of prior tasks. Although we don't evaluate backward transfer, we still include PNC as a representative example of prior continual learning methods. We also compare to the on-policy MAML method (Finn & Levine, 2018) in online mode (MAML+TL), where each iteration only meta-trains on the latest task. The on-policy MAML method is not designed for this setting, and therefore we expect it to perform poorly, but we include it for completeness.

**Meta-learning over stationary task distributions.** In our first set of experiments, we compare the methods on stationary task sequences. The results are presented in Figure 6. The plots show the average number of episodes needed to reach a success threshold on each of $40$ tasks, lower is better. In this evaluation protocol, once the fraction of successful trajectories exceeds a threshold, the algorithm moves on to the next task, and the goal is to solve all of the tasks as fast as possible. For additional details on how success is computed and the thresholds used see Appendix A. In these experiments, we can see that CoMPS solves all of the tasks the fastest, and in fact solves tasks *faster* as more tasks are experienced, indicating the benefits of meta-learning. This indicates the benefits of continual meta-learning, where each task enables the method to solve new tasks even more quickly. Without the option to collect on-policy data from each task MAML performs very poorly across all

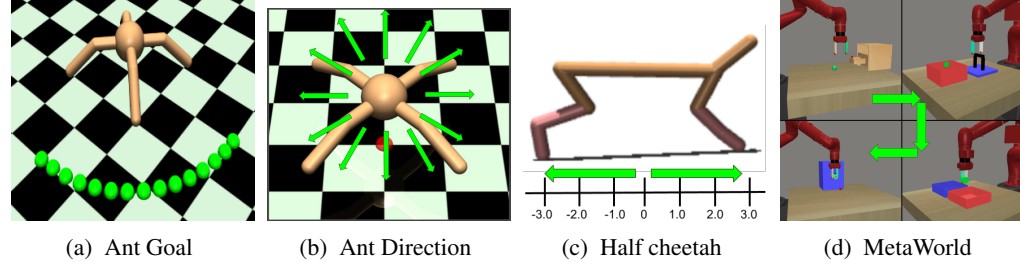

(a) Ant Goal      (b) Ant Direction      (c) Half cheetah      (d) MetaWorld

Figure 5: In **Ant Goal** the non-stationary task distribution selects the next goal location that is furthest from all previous locations: e.g., $\mathcal{T}_{0:i} = ((5,0), (-5,0), (0,5), \ldots)$. In **Ant Direction** the tasks start at $0°$ along the x-axis and rotate ccw $70°$. The **Half Cheetah** tasks start with low target velocity and alternate between larger $+-$ velocities: e.g., $\mathcal{T}_{0:i} = (0.5, -0.5, 1.0, \ldots)$. In **MetaWorld**, a set of goals is randomly choosen across a collection of environments including *hammar* and *close-drawer*.

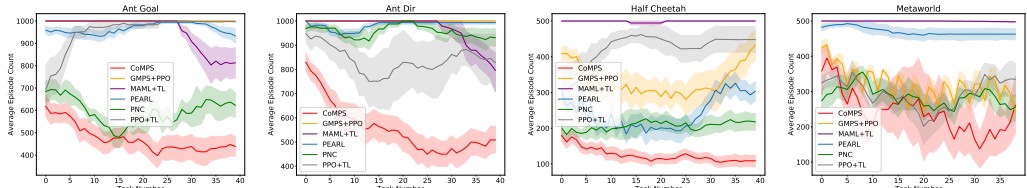

Figure 6: These figures show the average number of episodes needed to solve each new task after completing $i$ tasks (fewer is better). Results are computed over 6 sequences of 20 tasks, averaged over 6 random seeds with the standard error shown.

environments. Reaching the success threshold on the more challenging **MetaWorld** tasks is generally difficult, however, CoMPS still performs the best. In Appendix E we include additional results that show CoMPS receives higher reward on average for these experiments and in subsubsection E.1.1 we show the learning behavior of CoMPS is statistically different than other algorithms. The forgetting analysis in Appendix G shows CoMPS exhibits strong backwards transfer as well. The next paragraph provides fine-grained average reward analysis over non-stationary task distributions.

**Meta-learning over non-stationary task distributions.** In our second set of experiments, we evaluate all of the methods on non-stationary task sequences. The results of these comparisons are presented in Figure 7, which shows complete learning curves for each method over the task sequence (left), as well as a plot of the average performance on each task (right). The plot on the right is obtained by averaging within each task, and provides a clearer visualization of aggregate performance. The plots show that CoMPS attains the best performance across most task families, and the gains are particularly large on the higher-dimensional **Ant Direction** and **Ant Goal** tasks. Note that the decrease in performance on the **Half-Cheetah** task is due to the increasing difficulty of tasks later in the sequence, since the target velocities increase as more tasks are observed (see Figure 5 for details), but CoMPS still attains higher rewards than other methods. On **Ant Goal** and **Ant Direction**, the average performance of CoMPS increases as more tasks are seen, indicating that the meta-learning procedure accelerates the acquisition of new tasks. PEARL generally performs poorly across all tasks: although SAC is an off-policy algorithm, it is well known that such methods do not perform well when they are not allowed to gather any additional online data (as, for example, in the case of offline RL) (Kumar et al., 2019). PPO+TL and PNC provide strong baselines, but do not benefit from meta-learning as CoMPS does, and therefore their performance does not improve as much with more tasks. On the **MetaWorld** tasks, the best performing methods are CoMPS, PPO+TL and GMPS+PPO. In the plots on the left of Figure 7, we can see that CoMPS generally learns each task faster (as we expect due to meta-learning), but PPO+TL attains somewhat better final performance on most tasks. However, we can see in the Appendix, Figure 11 that CoMPS reaches these rewards in fewer interations compared to PPO+TL and GMPS+PPO.

**Ablation study.** We perform an ablation analysis to compare CoMPS to GMPS+PPO that does not use V-trace-based off-policy importance sampling. The results in Figure 6 show that GMPS+PPO can not make good use of the off-policy experience and, as a consequence, performs worse than CoMPS, especially on **Ant Goal** and **Ant Direction**. Additional ablation analysis showing the improvements from the importance sampling and training components of CoMPS are provided in Appendix H.

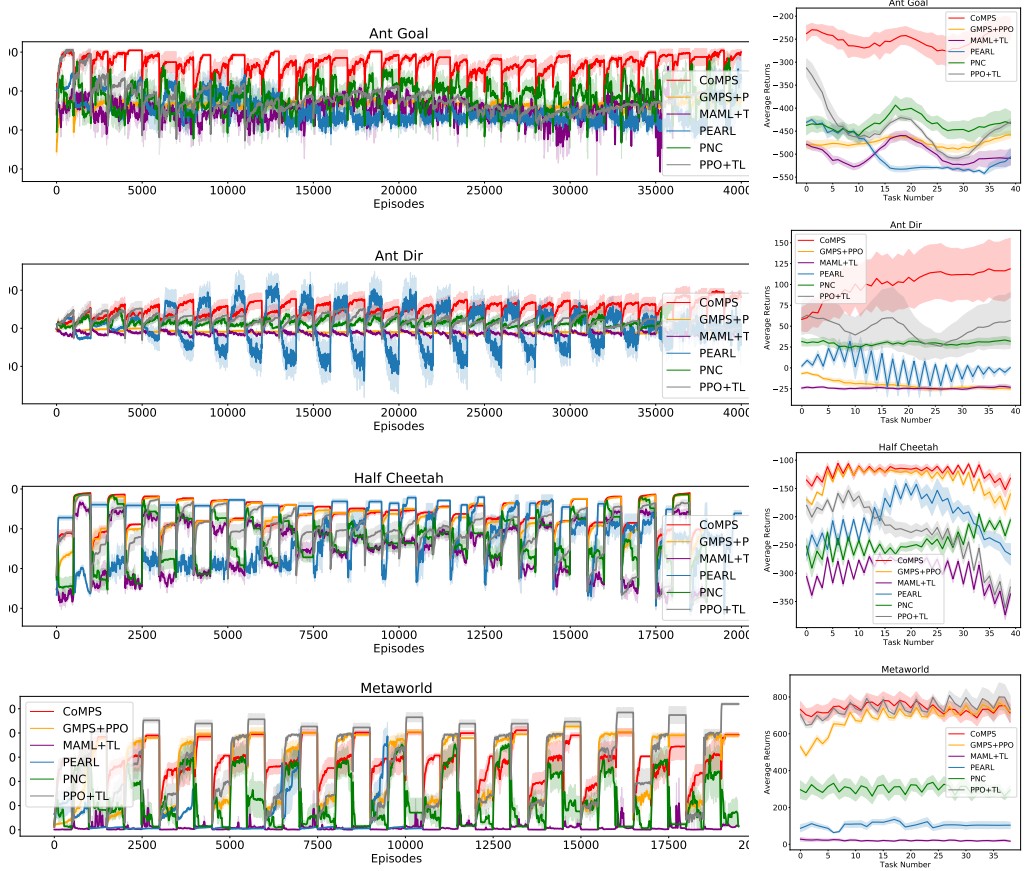

Figure 7: On the left are "lifelong" plots of rewards received for every episode of training over 40 tasks. The points are subsampled by averaging 5 points together to make one point in the plot. Results are averaged over 6 seeds, **Cheetah** and **MetaWorld** get 500 episodes and **Ant-Goal** and **Ant-Direction** get 1000 per task where each episode collects 5000 samples. The right plots show the average return across the episodes on each of the 40 individual tasks. CoMPS achieves higher average returns and improves its performance as more tasks are solved.

## 6 DISCUSSION

In this work, we proposed CoMPS, a new method for continual meta-reinforcement learning. Unlike standard meta-RL methods, CoMPS learns tasks one at a time, without the need to revisit prior tasks. Our experimental evaluation shows that CoMPS can acquire long task sequences more efficiently than prior methods, and can master each task more quickly. Crucially, the more tasks CoMPS has experienced, the faster it can acquire new tasks. At the core of CoMPS is a hybrid meta-RL approach that uses an off-policy importance-sampled inner loop policy gradient updated combined with a simple supervised outer loop objective based on imitating the best data from prior tasks produced by CoMPS itself. This provides for a simple and stable approach that can be readily applied to a wide range of tasks. CoMPS does have several limitations. Like all importance-sampled policy gradient methods, the variance of the importance weights can become large, necessitating clipping and other tricks. We found that including a V-trace off-policy value estimator for the baseline helps to mitigate this, providing better performance even for highly off-policy prior task data, but better gradient estimators could likely lead to better performance in the future. Additionally, CoMPS still requires prior data to be stored and does not provide for any mechanism to handle forgetting. While this is reasonable in some settings, an interesting direction for future work could be to develop a fully online method that does not require this. Since CoMPS does not require revisiting prior tasks, it can be a practical choice for real-world meta-reinforcement learning, and a particularly exciting direction for future work is to apply CoMPS to realistic lifelong learning scenarios for real-world applications, in domains such as robotics.

## 7 Ethics Statement

Current applications of robots in the real world require enormous amounts of supervision and instruction from scarce and skilled people. Our work can broaden the use of robotics by reducing steadily the time to learn new tasks, making them smarter as they operate. However, the method we have developed is a step in the direction of enabling robots to help provide essential services. Still, there is some risk in that skilled robots can be repurposed for other undesired tasks.

## 8 Reproducibility Statement

In an effort to make our work reproduceable we have provided extensive details for the method in the appendix including how our hyperparameter analysis was performed. We have also provided the code used for the experiments. This code includes instructions on how to reproduce the experiments.

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

## A EXPERIMENTAL DETAILS

Here we provide additional information on the task sequences and reward functions for each environment. The non-stationary task sequences were chosen to ensure neighboring tasks are different from each other. This was chosen so that we can focus on evaluating forward transfer to tasks that are dissimilar to previous tasks such that the agent will have to draw on old experience to perform well on each new task. For example, in the Ant Goal task, each successive task goal is 70 degrees more ccw compared to the last goal, ensuring a significant amount of adaptation is needed to solve the next task.

**Ant Goal.**

1. Goal description: With the agent starting at the origin $(x = 0, y = 0)$, goal locations are sampled 3 units away from the origin uniformly on a circle.

2. Non-stationary task sequence: We fixed twenty tasks, $[-171°, 9°, -162°, 18°, -153°, 27°, -144°, 36°, -135°, 45°, -126°, 54°, -117°, 63°, -108°, 72°, -99°, 81°, -90°, 90°, -81°, 99°, -72°, 108°, -63°, 117°, -54°, 126°, -45°, 135°, -36°, 144°, -27°, 153°, -18°, 162°, -9°, 171°, 0°, 180°]$, where each number reperesents the rotation of the goal's position, where for 0 the goal position of $(x = 3, y = 0)$ and 90 represents goal position of $(x = 0, y = 3)$. The stationary task distribution randomly shuffels this sequence.

3. Maximum Trajectory length $T^{max}$: We set $T^{max} = 200$, same as GMPS Mendonca et al. (2019).

4. Reward function: We modified the reward function in PEARL Rakelly et al. (2019), reducing the control cost portion of the reward by an order of magnitude. This is accomplished by multiplying the control value by 0.1.

**Ant Direction:**

1. Goal description: With the agent starting at the origin, goal directions, as vectors $(x, y)$, are sampled uniformly.

2. Non-stationary task sequence: We fixed forty tasks, $[-171°, 9°, -162°, 18°, -153°, 27°, -144°, 36°, -135°, 45°, -126°, 54°, -117°, 63°, -108°, 72°, -99°, 81°, -90°, 90°, -81°, 99°, -72°, 108°, -63°, 117°, -54°, 126°, -45°, 135°, -36°, 144°, -27°, 153°, -18°, 162°, -9°, 171°, 0°, 180°]$, where each number reperesents the degree of the goal's angle, e.g. 0 represents a goal that is aligned with the positive x-axis and 90 represents positive y-direction. For stationary distribution experiments the sequence is shuffled randomly.

3. Maximum Trajectory length $T^{max}$: We set $T^{max} = 200$, same as GMPS

4. Reward function: We modified the reward functions from the version of the environment used in PEARL by adding a scaling factor of 0.1 to the control cost.

**Half Cheetah:**

1. Goal description: The target velocities can be in the range $[-3, 3]$.

2. Non-stationary task sequence: We fixed forty tasks described by there goal velocity along the x-axis [-2.66, 0.14, -2.52, 0.28, -2.38, 0.42, -2.24, 0.56, -2.1, 0.7, -1.96, 0.84, -1.82, 0.98, -1.68, 1.12, -1.54, 1.26, -1.4, 1.4, -1.26, 1.54, -1.12, 1.68, -0.98, 1.82, -0.84, 1.96, -0.7, 2.1, -0.56, 2.24, -0.42, 2.38, -0.28, 2.52, -0.14, 2.66, 0.0, 2.8, ]. For the stationary task distribution the above sequence is randomly shuffled.

3. Maximum Trajectory length $T^{max}$: We set $T^{max} = 200$, same as GMPS Mendonca et al. (2019).

4. Reward function: There were no changes to the reward function from the environment used in PEARL Rakelly et al. (2019).

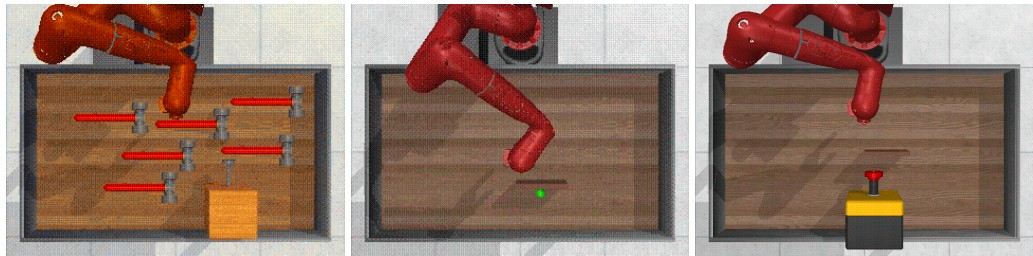

Figure 8: These figures show tasks use in the **MetaWorld** environment.

**MetaWorld:**

1. Goal description: We chose three environments from Metaworld: Push-Wall-V2, Button-Press-Wall-V2, and Hammer V-2. These environments were chosen based on their dissimilarity and difficulty measured by the level of *success* regular PPO could achieve on these tasks. For each task, we specify both the environment (Push-Wall-V2 (P), Button-Press-Wall-V2 (B), or Hammer V-2 (H)) and the goal index of that environment. The goal index controls additional variation in the tasks. For example, in the Hammer v-2 environment, shown in Figure 8 different goal tasks change the location of the hammer at the start of the episode.

2. Non-stationary task sequence: We fixed 39 tasks, [P-0, B-1, H-2, P-3, B-4, H-5, P-6, B-7, H-8, P-9, B-10, H-11, P-12, B-13, H-14, P-1P, B-1B, H-17, P-18, B-19, H-20, P-21, B-22, H-23, P-24, B-25, H-26, P-27, B-28, H-2H, P-30, B-31, H-32, P-33, B-34, H-35, P-36, B-37, H-38, ], where the letter on the left of '-' represents the environment ('H' for Hammer-V2, 'B' for Button-Press-Wall-V2, and 'P' for Push-wall-V2), while the number to the right of '-' specifies the goal index used in the environment. For the stationary task distribution, the above sequence is randomly shuffled depending on the random seed.

3. Maximum Trajectory length $T^{max}$: We set $T^{max} = 150$, same as the original Metaworld codebase Yu et al. (2020b).

4. Reward function: We used the same functions for the three environments as the original Metaworld codebase.

**Success Thresholds** For the experimental evaluation, we are interested in measuring how quickly an agent can solve tasks. Previous meta-learning environments (Yu et al., 2020b) have used a measure of *success* to determine the agent's progress on its tasks. These success measures are more robust to tasks that can have largely different reward scales. Each environment includes a success indicator that is 1 when the agent is within $\epsilon$ distance of the desired task goal and 0 otherwise. In this training version, the agent moves on to the next task when it has been told it has succeeded on the current one. This setting makes particular sense in the continual multi-task setting, where the goal is to do well on all tasks with minimal interaction. CoMPS uses these success thresholds as a means of forcing the agent to sufficiently solve tasks before moving on to new tasks. This helps increase the quality of the collected data. The maximum amount of time given for each task can be increased to ensure the agent has enough time to solve the task. Although, even if the prior task was not solved the data from that task is likely to be unique and add diversity to the meta training dataset which will help with learning more generalizable parameters (Gupta et al., 2018a).

1. Ant Goal: The agent is *successful* if the agent's position $(a_x, a_y)$ and the goal's position $(g_x, g_y)$ satisfy $(|a_x - g_x| + |a_y - g_y| < 0.5)$. We calculate the number of successful samples from a batch of trajectories and average the success across all samples. We found that agents trained through PPO training could reach a success rate of slightly over $0.3$. We therefore set the success rate *threshold* to $0.3$.

2. Ant Direction: The agent is successful if its unit direction vector $v_a$ and the unit goal direction vector $v_g$ satisfy $\|v_a - v_g\| < 0.4$ while at the same time, the agent's speed must be greater than 0.5 m/sec. We calculate the number of successful samples from a batch of trajectories and average the success across all samples. Through qualitative inspection of the agent, we found that expert behavior occurs at $50\%$ success. Therefore we use a success rate *threshold* of $0.5$.

3. Half-Cheetah Velocity: The half-cheetah agent is successful if the difference between its velocity $v_a$ and the goal velocity $v_g$ is less than 30% of the goal speed ($|v_a - v_g| < 0.3|v_g|$). We calculate the number of successful samples from a batch of trajectories and average the success across all samples (this is the same as Ant Goal and Ant Direction). We set the success rate threshold to $0.45$. The success threshold of $0.45$ was chosen by trial and error according to the ability of the learning agent. We found that the highest success PPO could attain, in terms of average success, across these tasks was $\sim 0.5$ and at $0.45$, the agent showcases expert skill on the task.

4. Metaworld: We use the built-in success function from the Metaworld codebase(Yu et al., 2020b). In the metaworld environments, we care about whether the agent has successfully performed the task during the episode. For this reason, the success value is computed differently than the other environments. We instead average the ratio of trajectories that contained at least one success. We found that agents trained with PPO can reach a success rate of over $0.2$ across most tasks while showcasing expert skills in the three environments. We set the success rate threshold to $0.2$. While the success thresholds that were used may appear low in terms of reward, we do find that their qualitative performance is good. This can be seen on the paper website, where visualizations of the policy results show that policies that pass the threshold qualitatively perform the correct behaviors. In most environments (Ant goal, Ant Dir, and Cheetah), it is impossible to be $100\%$ successful because the threshold is averaged over every timestep in a trajectory. With this, we can interpret the success threshold as meaning how fast the agent has reached the goal and maintained that goal. This means that for example, the success threshold of $0.3$ for Ant Goal may be the best any algorithm can do given the max episode length. A threshold of $0.3$ does not mean that the method fails $70\%$ of the time, but that it spends $30\%$ of the total time in the "success state" on average. This is why we used (single task) PPO, that is used in all methods in the paper, to determine the threshold – this amounts to saying that our threshold for success is the performance obtained by a good RL algorithm on only that task with plentiful data, is a reasonable "upper bound" for a few-shot method. Measuring success this way works well for these environments where, for example, Cheetah needs to maintain a desired velocity for as long as possible. However, for MetaWorld we used the same method from the MetaWorld paper that averages the success over all trajectories where success is $1$ for a trajectory if any step was a success. We do note that according to the meta world paper the best any flat RL algorithm could do at solving all tasks was achieving $\sim 30\%$ success. We were able to achieve a similar performance with PPO to the findings in MetaWorld of $25\%$. This and the lower performance of the meta-RL methods speak to the difficulty of the tasks. However, we do find that CoMPS achieves good qualitative results that can be seen on the webpage.

**Compute resources**  For all our experiments, we used different cloud compute resources. Across these resources, we trained using virtual machines with 8 CPUs and 16 GiB of memory.

## A.1 ENVIRONMENT DETAILS

We changed the control costs in the PEARL environments because we initially found that, in our continual setting, a naive PPO baseline would attain very poor results because it would focus too much on these costs, resulting in agents that stood still and did not perform the task. After the reduction in control costs, naive PPO demonstrated a substantial improvement in solving each task across the PEARL environments. This change also makes the PEARL environment similar to the MetaWorld environments, which do not use any control torque costs. However, for completeness, we reran all our experiments in the original environments from the PEARL paper (without any changes to the reward functions). We find that generally, the trends are similar, with the exception that PPO+TL and CoMPS perform similarly on the Ant-Dir in the non-stationary task distribution case.

## B COMPARISON TRAINNING DETAILS

Here we provide training details for the algorithms we compare.

**PPO+TL:**   This algorithm uses a version of PPO that many methods share in this work. This agent trains on each task using only the stochastic gradient objective outlined for PPO. Unlike CoMPS, between tasks PPO+TL does not perform additional meta-learning and instead transfers the network parameters learned this far directly to the next task.

**PEARL:**   The comparison to PEARL uses the provided codebase released with the paper (Rakelly et al., 2019). We updated the PEARL code to change the tasks the PEARL agent trains over. Instead of the original $\sim 130$ tasks used for training, we refine the agent to train on one task at a time according to the task distributions outlined in the paper. The agent collects 5000 simulation steps between evaluations.

**PNC:**   The PNC algorithm consists of two phases similar to CoMPS. One phase is normal RL which we train using the same parameters and version of PPO that we used for both CoMPS and PPO+TL. The second phase is a *distilation* phase that uses *elastic weight consolidation* to train a different head of the network to imitate the behavior of the current policy on that task while reducing the forgetfulness via a constraint of the change in the output distribution. This distillation phase needs on-policy data; therefore, we devote the last $\%5$ of training to this distillation phase in our experiments. PNC also uses a different network structure than all other methods in the analysis. For PNC the network consists of a policy and a knowledge base with the additional output head for imitation/distillation. Both networks have hidden layers of size $[128, 128]$ and according to (Schwarz et al., 2018) the layers in the knowledge base are connected to the layers of the policy.

## C   RL Trianing Details

At the end of RL training, we split the data into two portions. These are the $\mathcal{D}_k^*$ and $\mathcal{D}_k$ collections. The number of trajectories of $\mathcal{D}_k^*$ depends on $T^{max}$ for each task, but we fix the batch size to be 5000 samples. We save a portion of the data the RL agent generated to reduce memory costs. We keep a portion of the min of $N$ episodes on RL or the $M$ episodes needed to reach the success threshold. Mathematically, we sample $k$ episodes of data where $k = \min(M, N \cdot 0.05)$. Then we randomly sample 100 trajectories from the $k$ episodes of data. When $M$ is large, this processing avoids keeping around too much data that may lead to memory issues on the computer.

## D   CoMPS Details

For the RL part of CoMPS, GMPS+PPO, PPO+TL, and PNC we explored the best hyperparameters by evaluating PPO on each task in parallel starting from a randomly initialized policy. We searched for the best values for the *learning_rate*, *stochastic policy variance*, *kl_clipping_term* (for PPO), and network shape that would receive the highest reward over the individual tasks described in the paper. For the meta-step (M) that CoMPS and GMPS+PPO share we tuned those algorithms using a similar process. We searched for the best parameter values of *learning_rate*, the number of training updates per batch of data, alpha, batch size, and the number of total iterations or after each new task. We selected the parameters that achieved the highest stable rewards during training. We used the same values for both CoMPS and GMPS+PPO. For PNC we also performed hyperparameter tuning over the learning rate, the number of consolidation/compress steps, and EWC weight. For PEARL we performed hyperparameter tuning over the learning rate, batch size and the number of gradient steps. We believe that this procedure provides the most favorable possible hyperparameters for the prior methods and baselines.

We used grid search to find the best parameter settings. The search would include 3-4 different parameters settings for each hyperparameter. For example, here is the json code that lists the grid of parameters searched for PPO.

```
{
    "ppo_learning_rate": [0.01, 5e−3, 0.001, 0.0001],
    "hidden_sizes": [[128, 128], [128, 64], [64, 64]],
    "init_std": [0.3, 0.5, 0.7],
}
```

After this search, we used these same parameters for all methods that used PPO and only searched method-specific hyperparameters. For example, here is the list of hyperparameters evaluated for CoMPS:

```
{
    "adam_steps": [50, 100, 250, 500],
    "meta_itr": [20, 50, 100],
    "meta_step_size": [0.005, 0.001, 0.0005],
}
```

The same process is used across all methods and baselines. The only difference is the additional hyperparameters that may be available to different algorithms. We will include these details in the final version of the paper. The score function that is used to rate the search parameters is the average reward over all tasks and environments. We endeavor to use the same budget for each method by allowing for an equal number of randomly seeded experiments. An average of 36 combinations of parameters to test each with 6 random seeds leads to 216 simulations per method. These simulations were run in parallel on cloud computing technology (AWS/GCP/Azure/slurm). We will include these details on the number of simulations used and the hardware it was run on in the final version of the paper.

We attempted to select parameters based on only a single task, however, those parameters did not generalize well to all tasks and environments. We did find that this simple grid search over hyperparameters generally provided better performance for all baselines as compared to their default parameters, and therefore in the paper we used the best settings for all baselines to ensure a fair comparison.

Using the above analysis we found a learning rate of 0.005 for each CoMPS meta-learning step, with a total of 25 training steps. For all experiments, we use a two-layered hidden network of sizes $[128, 64]$. We use a single inner gradient step for all tasks with inner learning rates shown below (Algorithm 1 line 5). For the outer step of CoMPS, we perform imitation learning on $\mathcal{D}_k^*$ for 5 steps on Algorithm 1 line 7.

| Environment | Inner Learning Rate |
|---|---|
| Ant Goal | 0.005 |
| Ant Direction | 0.005 |
| Half-Cheetah Velocity | 0.005 |
| Metaworld | 0.0025 |

Table 1: CoMPS Hyperparameters

# E  ADDITIONAL RESULTS

Additional training results across environments.

## E.1  STATIONARY TASK DISTRIBUTION

In Section 5 we performed experiments to evaluate the performance of CoMPS compared to other methods in terms of learning speed. This performance is measured as the number of learning episodes needed for an agent to *successfully* solve the task. These results can be found in Figure 6. Here we include additional results and metrics for evaluation in addition to these results to get a broader view of the findings. In Figure 9 we show the average return received for the same experiments as in Figure 6. In these results that use uniformly random task distributions, we can see that CoMPS achieves higher average reward while completing the tasks. The CoMPS also improves its ability to achieve high rewards quickly as more tasks are solved. This performance improvement can also be seen in Figure 12(top) that shows the return averaged over tasks, where we average the learning graphs together for each task.

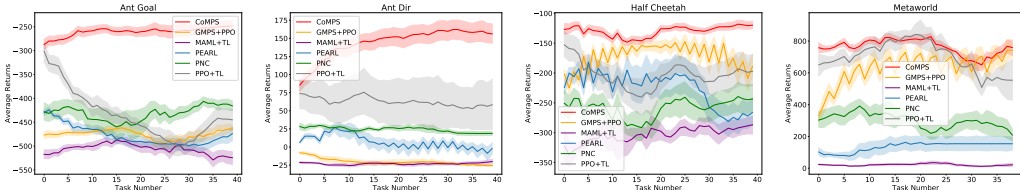

Figure 9: These figures show the average return for each new task. On average CoMPS achieve the highest average return while training over an entire task at a time. Results are computed over 6 sequences of 40 tasks, averaged over 6 random seeds.

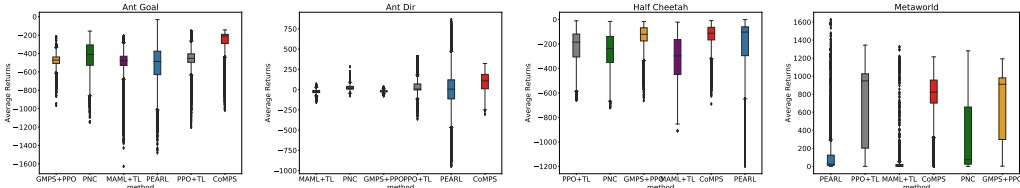

Figure 10: These figures show the average return achieved during the $RL$ phase over all tasks. This shows average return the agent achieves if any episode is picked at random from any task. Still, CoMPS performs better than other methods considering this analysis. Results are averaged over 6 sequences of 40 tasks, and 6 random seeds.

### E.1.1 STATISTICAL SIGNIFICANCE

We perform a two-sided independent t-test on the learning data from the stationary task distribution experiments using a bootstrap of 30, 000 over 6 randomly seeded trials. This test provides information on whether the baseline results appear to be from the same distribution as CoMPS. Our findings are shown in Table 2. This analysis shows that the distribution of average number of episodes needed to complete a task for all algorithms is not the same as CoMPS. This analysis is suggested in Henderson et al. (2017) to provide futher data on the statistical significance of the results.

| Alg | Ant Goal | Cheetah | Ant Direction | MetaWorld |
|---|---|---|---|---|
| PEARL | 1.85E-11 | 0.0002371241398 | 7.36E-09 | 2.26E-13 |
| PPO+TL | 4.54E-11 | 2.66E-05 | 0.01990761802 | 0.006441451648 |
| MAML+TL | 1.12E-11 | 4.17E-18 | 1.87E-08 | 0.007026979433 |
| PNC | 0.01833026502 | 0.002410406845 | 6.35E-09 | 1.26E-10 |

Table 2: p-values from a two-sided independent t-test (CoMPS, Alg) performed over the stationary task distribution experimental data.

### E.2 NON-STATIONARY TASK DISTRIBUTION

In Figure 7(left), we show the average reward of each method for every episode of learning over 40 tasks from a non-stationary distribution. On the right of Figure 7 we show the average reward achieved over the same 40 tasks. To provide more aggregate data, we also compute the average reward achieved over all tasks. We provide this information in Figure 10. In this data, we can see that CoMPS obtains a median average return that is higher than all other methods. PEARL, in particular, exhibits a high variance in rewards across tasks, which can also be seen in Figure 7(left). GMPS+PPO, which does not include the additional off-policy importance weighting corrections used in CoMPS, achieves some of the lowest returns across all methods. This further illustrates the importance of the off-policy corrections in CoMPS. We also look at performance per episode averaged over tasks in Figure 12(top). The reason these curves are smooth with low variance is because of the approximately 100 samples being averaged over 6 x number of tasks = 240. CoMPS accumulates the largest area under the curve compared to all other methods in the stationary and non-stationary case. These results also show that CoMPS achieves performance improvments in early episodes across tasks compared to all other methods.

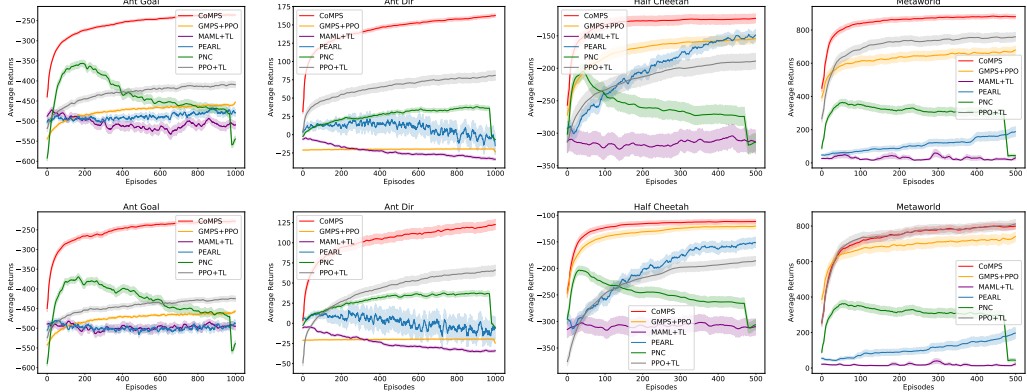

Figure 11: Similar to Figure 6, these figures show the average number of learning iterations it takes for each method to reach success *on the non-stationary tasks.* In this case we can see that CoMPS solves most tasks faster than other methods.

Figure 12: These figures show the average return achieved for each episode in the $RL$ phase over all tasks. The top row is for the stationary task distribution and the bottom for the non-stationary task distribution. These graphs show that CoMPS is achieves the highest average return over all other methods when performance is averaged over tasks. Results are averaged over 6 sequences of 40 tasks, and 6 random seeds.

### E.2.1 STATISTICAL SIGNIFICANCE

We perform a two-sided independent t-test on the learning data from the non-stationary task distribution experiments using a bootstrap of 30,000 over 6 randomly seeded trials. This test provides information on whether the baseline results appear to be from the same distribution as CoMPS. Our findings are shown in Table 3. This analysis shows that the distribution of average reward for most algorithms is not the same as CoMPS. In the **MetaWorld** and **Ant Direction** environments PPO+TL is very similar to CoMPS and somewhat similar to CoMPS respectively.

| Alg | Ant Goal | Cheetah | Ant Direction | MetaWorld |
|---|---|---|---|---|
| PPO+TL | 0.0003071245518 | 7.32E-08 | 0.2102584056 | 0.9973906496 |
| PNC | 0.0002119448748 | 3.93E-08 | 0.07235237317 | 1.49E-05 |
| MAML+TL | 0.0001044653709 | 7.91E-08 | 0.01082454019 | 8.97E-06 |
| PEARL | 0.0001308992933 | 1.39E-06 | 0.02516455006 | 9.79E-06 |

Table 3: p-values from two-sided independent t-test (CoMPS, Alg) performed over the non-stationary task distribution experimental data.

### E.3 VIDEO RESULTS

To see videos of the learned policies, see https://sites.google.com/view/compspaper/home

## F FURTHER USE CASES

We believe that there are many settings in robotics and other domains where revisiting previous tasks is an issue. For example, imagine a robot at a recycling plant that has to disassemble electronics: each item of electronics might be different, and once it has been disassembled, it cannot be disassembled

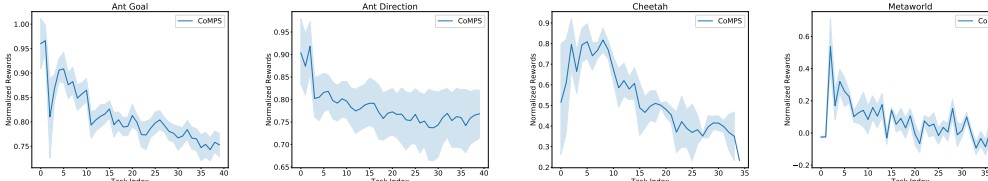

Figure 13: These figures show the backwards transfer for CoMPS averaged over prior tasks. The y-axis if the *normalized reward* where 1 means the method retained the original performance on the task.

again. So there is a sequence of tasks, prior tasks cannot be revisited, and it is desirable to accelerate all future tasks. Similarly, imagine a house cleaning service performed by a robot that puts away objects – once the robot has cleaned one house, it goes to a different one. Revisiting houses in a round-robin fashion would be impractical and unnecessary because the houses will already be clean.

## G  BACKWARD TRANSFER

In Figure 13 we show the backwards transfer of CoMPS. We plot the *normalized reward* $= \frac{c_{0,...,k} - a_{0,...,k}}{b_{0,...,k} - a_{0,...,k}}$, where $a_{0,...,k}$ is the average reward over the *first* iteration of learning for the previous $k$ tasks (representing the lowest reward the agent recieved), $b_{0,...,k}$ is the average reward on the *last* iteration (representing the highest reward the agent recieved). Finally, $c_{0,...,k}$ is the average reward the CoMPS agent recieves when evaluated over the prior $k$ tasks. When the *normalized reward* is 1 there is no loss in performance on the tasks, when the value is 0 the method has forgotten how to perform the task. We can see from these plots that CoMPS exhibits strong backwards transfer on the **Ant Direction**, **Ant Goal**, and **Cheetah** environments. For **Ant Goal** and **Ant Direction** backwards transfer appears to stabilize, indicating that learning new tasks may not affect performance on old tasks. This observation aligns with the properties of meta-learning and MAML such that adding more tasks generally increases adaptation performance. The reducing performance on **Cheetah** is likely related to the increasing difficulty of tasks. CoMPS demonstrates backwards transfer on **Metaworld** initially; however, this reduces as more tasks are seen, again indicating the difficulty of discovering generalizable structure across the diverse **Metaworld** tasks.

## H  COMPS ABLATION

Here, we perform additional analysis on the components of CoMPS and how they affect the performance of the overall method. GMPS performs very poorly across all tasks. The addition of PPO (GMPS+PPO) increases performance on **Cheetah** and **Metaworld** but the relative performance is still poor. A significant improvement is given with the introduction of off-policy importance sampling (GMPS + PPO + IS). As described in the paper, the importance sampling allows CoMPS to compute unbiased gradient information from a large collection of off-policy data. We also investigate storing additional *skilled data* at the end of learning each new task (GMPS with IS + x2 the skilled data). The addition of more *skilled data* does not generally improve the performance of CoMPS. Following a pattern similar to FTML Finn et al. (2019), CoMPS initializes meta-RL training after each task from the policy trained on the previous task, across environments this improves learning speed (CoMPS/ours (GMPS + PPO + IS + FTML)). In particular, the addition of FTML-based training decreases the variation in performance. It results in a smoother reduction in the time used by CoMPS to solve new tasks.

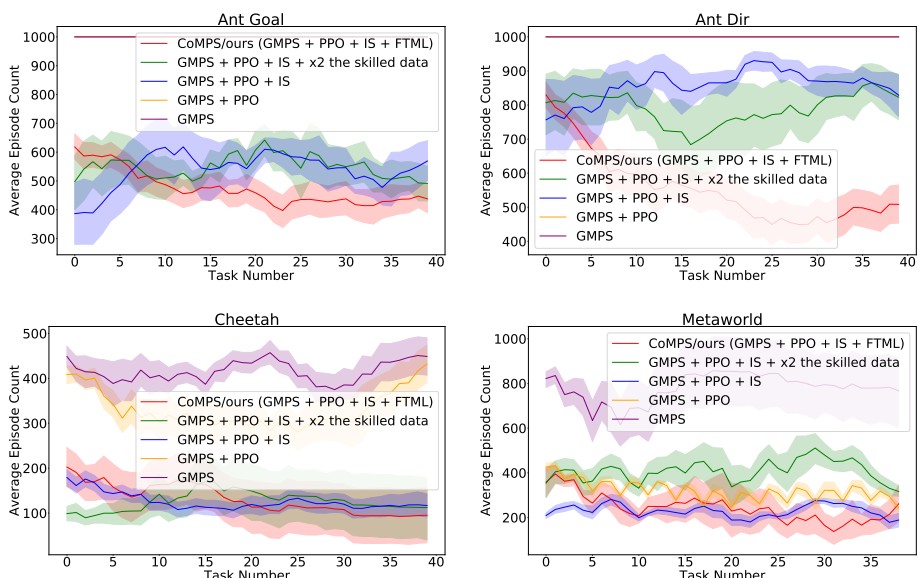

Figure 14: These figures show performance of variations of CoMPS. As we remove features from CoMPS we can see the performace reduce considerably.

