# OpenReview forum: "CoMPS: Continual Meta Policy Search"
_ICLR.cc/2022/Conference — ICLR 2022 Poster_

### Official Review · Reviewer_1xJQ · 2021-10-21

**Correctness:** 3
**Technical Novelty And Significance:** 2
**Empirical Novelty And Significance:** 2
**Recommendation:** 3
**Confidence:** 4

**Main Review:**

The main novelty of this work is the proposed continual meta RL setup and the design of the experiments. Regarding the algorithm, I find it to be reasonable application of existing ideas to this setup.
To give a few examples:

The ablation study indicates that the method performs better by using an existing algorithm for doing off-policy learning. While this is nice to know, it is not very surprising, and we can learn very little from looking at figure 6 about that. A more carful investigation of this question will include an experiment that is centered on the off policy questions in this setup, will report some more statistics (e.g., the IS weigths as the tasks change) and will compare to more baselines.

The baselines that were chosen are complete methods, that were not specifically designed for this new setup and are not high performing in it for that reason. A much more interesting comparison would be a carful ablation analysis on top of COMPS showing which components make it superior and when. For example, I don't learn a lot from the comparison to PEARL as a whole, and I would appreciate a comparison that uses COMPS with SAC as the RL agent. I am interested to know what is needed in this setup and why, I am less interested why papers that were designed for different setups are being outperformed (while it is good to know that this is the case).

To summarize, I feel that the paper will have to make stronger algorithmic contributions and more carful ablation study for it to get accepted to ICLR. In this form, I would recommend rejecting it.

**Summary Of The Paper:**

The paper propose a continual meta-learning setup where the agent is faced with a sequence of tasks and has to adapt quickly to new tasks based on the experience it gathered in previous tasks. In addition, at future iterations the agent can only interact with the previous tasks by doing off-policy learning on the data it collected during its interaction with these tasks and cannot further interact with these previous tasks. The authors then propose an algorithm that combines RL on the current task with offline meta RL on the previous tasks. The authors then construct setups that fit this problem setup, evaluate their method on it and compare to previous baselines.

**Summary Of The Review:**

Interesting meta RL setup and nice experiment construction. At the same time, the algorithmic novelty is questionable and the ablation study is not convincing enough.

---

> ### Author Response · Authors · 2021-11-13
> **Author Response**
>
> Q1: I feel that the paper will have to make stronger algorithmic contributions and more carful ablation study for it to get accepted to ICLR.
>
> A1: We would first emphasize that our main contribution is the continual meta-RL problem formulation and (to our knowledge) the first empirical demonstration that continual meta-RL can accelerate acquisition of new tasks without revisiting old tasks. We believe this is significant, because this setting is common in the real world (e.g., a robot can’t easily and instantly revisit past environments), and prior work has not shown that meta-learning accelerates continual acquisition of tasks in this setting. Novel empirical findings are highly relevant to the ICLR community, even when the methodology to obtain these findings does not contain particularly complex mathematics or drastically new algorithmic components, and much of the research in deep learning and RL falls into this category.
>
> In terms of the method: While our method does build on prior methods, including GMPS and V-Trace, the experimental results show that the differences are very important: In our experiments, CoMPS performs better than GMPS in the continual meta-reinforcement learning setting in every one of the experiments. Indeed, GMPS often performs worse in this setting than the comparatively naive PPO baseline, suggesting that without the proposed changes, it cannot tackle these tasks. Many papers that propose new algorithms in practice make comparatively small modifications to prior work if we look closely (e.g., TRPO is mechanically a small modification of natural gradient, DDPG is a small modification of NFQCA, even DQN is a small modification of existing Q-learning methods at the time). The value of these works is in showing that such changes are important for performance.
>
> Q1: include an experiment that is centered on the off policy questions in this setup, will report some more statistics (e.g., the IS weigths as the tasks change) and will compare to more baselines.
>
> A1: We are running this experiment now. We are also running additional experiments to study the affected of storing different amounts of data for the off-policy dataset and the skilled dataset and the change in performance when meta-RL is not initialized from the RL policy from the prior task. We will report back on these results soon.
>
> Q2: I don't learn a lot from the comparison to PEARL as a whole, and I would appreciate a comparison that uses COMPS with SAC as the RL agent.
>
> A2: This is an interesting idea however it is not clear how this would work. The MAML formulation uses policy gradients to train on-policy. This makes it difficult to train a meta-policy that can be properly used to initialize an SAC-based agent, and we are not aware of any prior method that uses MAML-style updates with SAC. This limitation is also a reason prior methods, such as GMPS, are on-policy, and therefore cannot be directly applied to the problem of continual meta-RL.

---

> ### Author Response · Authors · 2021-11-18
> **Feedback**
>
> Hello reviewer 1xJQ, we would be grateful if you can confirm whether our responses have addressed your concerns, and let us know if any issues remain. To recap our response, we:
>
> - Explained in more detail the importance of the contributions and how the amount of novelty and contribution is similar to prior published papers.
> - Commented on how it is difficult to combine SAC and MAML, therefore we needed to develop CoMPS for continual meta-reinforcement learning.
>
> We have also included a new backwards transfer analysis of CoMPS in Section G.

---

> > ### Comment · Reviewer_1xJQ · 2021-11-18
> > **Response to author feedback.**
> >
> > I thank the authors for their clarifications. However, I still find the novelty of this work to be limited for the reasons I mentioned in my review and the authors rebuttal did not convince me otherwise. In particular, I find the comparisons with TRPO DDPG and DQN in terms of novelty to be exaggerated.

---

> > > ### Author Response · Authors · 2021-11-19
> > > **Continued discussion**
> > >
> > > We would emphasize that the main contribution of our paper is not the particular set of modifications needed to devise a continual meta-learning method from the parts of existing algorithms, but rather the formulation of the continual meta-learning problem and (to our knowledge) the first method for tackling this problem that demonstrates that continual experience on past tasks accelerates the acquisition of new tasks. We show that prior methods generally do not enable this. It is possible to make our algorithm more complicated just for the sake of making it appear more novel, but this would be pointless: if a simpler method that combines existing components (albeit in a novel way) makes meaningful progress on a new problem that prior methods generally do not solve effectively, isn't this better than trying to cook up a more complex method just for the sake of making it more complex? The novelty of a result is not based on the edit distance between the new algorithm and past algorithms, but on whether it enables something new and desirable, and we believe that our experimental results back that up. By analogy, FTML [1], which enables online meta-learning in the supervised setting (but does not apply to the RL setting) is arguably even more closely based on past algorithms, since it just directly applies MAML to the online setting, but nonetheless was a successful result in the meta-learning literature.
> > >
> > > - [1] Finn, Chelsea, Aravind Rajeswaran, Sham Kakade, and Sergey Levine. 2019. “Online Meta-Learning.” 36th International Conference on Machine Learning, ICML 2019 2019-June: 3398–3410.

---

> > > ### Author Response · Authors · 2021-11-25
> > > **Feedback appreciated**
> > >
> > > Dear Reviewer, We hope that you've had a chance to read our response. We would really appreciate a reply as to whether our response and clarifications have addressed the issues raised in your review, or whether there is anything else we can address.

---

### Official Review · Reviewer_cHnK · 2021-11-01

**Correctness:** 3
**Technical Novelty And Significance:** 2
**Empirical Novelty And Significance:** 2
**Recommendation:** 5
**Confidence:** 2

**Main Review:**

strengths:
- this paper considers a more realistic and difficult scenario, continual Meta-RL learning, and proposed an extended meta-rl framework that adapts this setting;
- the experiments are sufficient and the performance improvment is significant;
- the writing is good and key details are clearly described.

weakness:
- This paper can be viewed as a combination of several existing techiniques. In particularly, this paper claims that it addressed a type of continual learning problem but did not consider one of the main characteristic of contual learning - i.e., the forgetting issue. Hence a reasonable baseline to use the past experience is the offline RL learning instead of consider it as a meta learning problem. Overall，it looks to me that the whole method is a kind of  MAML + off-policy RL + online RL.

- though the paper claims that this is the first to  formulate and address the continual meta-learning problem, this problem setting has already been discussed before, some work can be referred to:
[1] Online Meta Learning. Chelsea Finn, et.al.
[2] Continuous adaptation via meta-learning in nonstationary and competitive environments. Maruan Al-Shedivat, et.al.

some questions and typos:
1. the inner optimization for each sequential task is actually separate, so why not use a recurrent neural networks structure to avoid the forgetting by utilizing the historical information?
2. the subscripts of both summation and quadrature symbols in Eq.2 have problems.
3. the caption of Figure.7  is wrongly written as '... the average average return across the episodes ...'.

**Summary Of The Paper:**

This paper considers  the challenges in sequential multi-task learning, in which the agent will visit all the tasks in a sequence, and will not revisit the previously learned task. This paper proposed a continual meta-RL framework, named Contunual Meta-Policy Search (CoMPS), so as to extend the traditional meta RL framework, GMPS(Guided Meta Policy Search) to a continual Meta-RL setting. Different with GMPS, CoMPS firstly adopts self-imitation learning as the outer optimization instead of a true imitation learning and secondly adopts an importance sampling-based policy gradient for off-policy inner optimization, which is suitable for the continual Meta-RL. The empirical results show that CoMPS can outperform the prior continual learning or meta learning methods on both stationary and non-stationary task distributions.

**Summary Of The Review:**

The issues considered and the solutions proposed in this paper sound reasonable, and the proposed method does achieve the better performance. But some details of the whole pipeline need more justification, and the optimization between sequential tasks seems to be independent. I'm also not sure that this is able to address the issue about catastrophic forgetting.

---

> ### Author Response · Authors · 2021-11-13
> **Author Response**
>
> We agree that dealing with forgetting (stability) is an important component of the continual learning problem. However, the concept of forward transfer (plasticity) is also very important to the community. This is especially true for areas of the community where large datasets are available and the focus is on how to learn generalization skills to adapt to new tasks. This is true for the robotics example described in the introduction, where a robot can collect and store experience from visiting and cleaning many homes. The robot will have to clean a new house the next day and it will be able to clean the new house the fastest if it can transfer forward as much knowledge as possible to the new task. Also, after the robot has cleaned the old homes those tasks no longer exist and devoting resources to remember how to perform tasks that may never exist again seems counterproductive. CoMPS is designed to address the challenge of learning new tasks quickly (plasticity). CoMPS must still retain prior data, and we do not claim to fix this problem, though it may be important to tackle in future work. We have however shown across multiple environments that CoMPS provides an improvement over prior methods in terms of plasticity, and we believe this is significant. We appreciate the reviewers’ concerns and we will make it clear much earlier in the paper that our work focuses on forward transfer, that retaining prior data is one of its limitations, and that we leave a full treatment of the forgetting problem for future work. However, we believe this scoping is reasonable, since in many settings it is possible to retain prior data, as is done in several rehearsal-based prior works on supervised online learning [1, 2, 3, 4].
>
> In terms of the method: While our method does build on prior methods, including GMPS and V-Trace, the experimental results show that the differences are very important: In our experiments, CoMPS performs better than GMPS in the continual meta-reinforcement learning setting in every one of the experiments. Indeed, GMPS often performs worse in this setting than the comparatively naive PPO baseline, suggesting that without the proposed changes, it cannot tackle these tasks. Many papers that propose new algorithms in practice make comparatively small modifications to prior work if we look closely (e.g., TRPO is mechanically a small modification of natural gradient, DDPG is a small modification of NFQCA, even DQN is a small modification of existing Q-learning methods at the time). The value of these works is in showing that such changes are important for performance.
>
> Q1: paper claims that this is the first to formulate and address the continual meta-learning problem.
>
> A1: We claim that we are the first to address the continual meta-reinforcement learning problem. This is true. The provided referenced works perform research on the problem of continual meta-learning for supervised learning problems, such as image classification. Other prior methods do not support off-policy training and assume smooth Markov structure between task transitions. CoMPS does not make these assumptions and has still been shown to improve learning speed from meta-reinforcement learning as more tasks are solved. We will add similar text to the paper to make these distinctions more clear.
>
> Q2: the inner optimization for each sequential task is actually separate, so why not use a recurrent neural networks structure to avoid the forgetting by utilizing the historical information?
>
> A2: We are not exactly sure what is being asked from this comment. It seems you are suggesting to encode the agent’s history into an RNN. If we look at PEARL under the right light it can be considered to capture essentially this general idea. To bring PEARL closer to your suggestion the PEARL network could be replaced with an RNN, though we are not aware of any prior continual meta-RL method that does this. RL2 does this, but it is an on-policy method, and therefore would be difficult to apply to this setting. Even this potential version of PEARL should suffer from the same extrapolation issues as the version evaluated in our paper. CoMPS does use an independent meta-RL optimization between tasks is a feature of CoMPS that allows CoMPS to meta-RL train using offline data. This makes the overall learning process more efficient than prior methods, including PEARL.

---

> > ### Comment · Reviewer_cHnK · 2021-11-30
> > **Response to Author response**
> >
> > I appreciate the authors' response. However, I think that the  robotics example raised is not very representative - in my opinion, clearing a new room cannot be thought of as a new task except that the room is in a different house. In other words, It seems to me that the skill transfered by the agent of this paper can only be applied in a very similar environment to the previous encountered ones, and in such cases, I agree that there is no forgetting problem to be handled. Otherwise, I cannot image why the proposed method could transfer smoothly all its learnt knowledge or skill from previous tasks to a completely new task (with different transition model and reward function). This latter point is not verified in the experimental section as well.
> >
> > In addition, as pointed out by R1, the paper  emphasized too much on what it can do, but not on why it has such capability and the original key idea behind to achieve that.
> >
> > Due to the above reasons, I would keep my score unchanged.

---

> > > ### Author Response · Authors · 2021-11-30
> > > **Continued discussion**
> > >
> > > Hello,
> > >
> > > Thank you for your additional discussion on the paper.
> > >
> > > As you say tasks new tasks can have different reward functions or transition models. Many meta-learning papers use these differences to construct different tasks for meta-training [MAML, GMPS, PEARL, MetaWorld]. For example, Metaworld includes environments such as hammering a nail where for each task the location of the nail is different, similar to going into different rooms that have different nail locations. The reward function may stay the same but the dynamics model is not exactly the same in each room, such that if the agent tries to hammer in the same location that worked in the previous room it will not find a nail in the new room. In addition, we can consider an electronics recycling task where a robot needs to disassemble items it is given at a recycling plant. After the robot is done disassembling an item that item can not be disassembled again. If the robot can learn the similar structure between objects it can learn how to disassemble each new object faster. We discuss this example as well in Appendix F. However, we believe CoMPS also has more immediate value to the continual learning as a new algorithm based on meta-reinforcement learning and to the meta-RL community as it is introducing a method and problem setting for meta-RL that has not been presented before but will assist the continual learning community.
> > >
> > > CoMPS may generalize better to tasks that are similar to prior tasks. This is true, and the more similar the new task is to prior tasks the faster CoMPS may generalize. However, we also show that the generalization ability of CoMPS to new tasks improves as more tasks are solved (Figure 6). In particular, for the non-stationary task distribution where tasks are chosen to be different from prior tasks (Figure 7). This shows that CoMPS can increase learning efficiency even when new tasks are reasonably different than old tasks. We will include a more nuanced discussion in the final text on the generalization capabilities of CoMPS.
> > >
> > > It is true that if there is no similarity between new tasks and prior tasks then there will be no transfer, this is not unique to CoMPS. All meta-learning algorithms will be limited in this case. However, because (1) CoMPS is based on MAML and (2) CoMPS can train using off-policy data, if there exists some overlap between the stored data from prior tasks and the new task it is possible for CoMPS to improve learning efficiency on the new task. The key reason why CoMPS performs as well as it does is because of the combination of (1) and (2) and our experiments show that this is a strong combination for the continual meta-RL setting over prior methods. The new ablations in Appendix H also make this point more clear. We will update the final text in the paper to make these points on why CoMPS has the ability to improve over prior methods more clear.

---

> ### Author Response · Authors · 2021-11-13
> **Author Response Continued**
>
>
> Q3: Typos
>
> A3: These will be fixed. Thank you.
>
> - [1] Isele, David, and Akansel Cosgun. 2018. “Selective Experience Replay for Lifelong Learning.” AAAI 32.
> - [2] Riemer, Matthew, Ignacio Cases, Robert Ajemian, Miao Liu, Irina Rish, Yuhai Tu, and Gerald Tesauro. 2018. “Learning to Learn without Forgetting by Maximizing Transfer and Minimizing Interference.”
> - [3] Rolnick, David, Arun Ahuja, Jonathan Schwarz, Timothy P. Lillicrap, and Greg Wayne. 2018. “Experience Replay for Continual Learning.”
> - [4] Atkinson, Craig, Brendan McCane, Lech Szymanski, and Anthony Robins. 2021. “Pseudo-Rehearsal: Achieving Deep Reinforcement Learning without Catastrophic Forgetting.” Neurocomputing 428 (March): 291–307.

---

> ### Author Response · Authors · 2021-11-18
> **Feedback**
>
> Hello reviewer cHnK, we would be grateful if you can confirm whether our responses have addressed your concerns, and let us know if any issues remain. To recap our response, we:
>
> - Discussed the contributions of CoMPS in comparison to prior work
> - Clearly noted that CoMPS is the first continual meta-**reinforcement** learning algorithm.
> - Gave more details on the CoMPS algorithm.
>
> We have also included a new backwards transfer analysis of CoMPS in Section G.

---

> ### Author Response · Authors · 2021-11-25
> **Feedback followup**
>
> Dear Reviewer, We hope that you've had a chance to read our response. We would really appreciate a reply as to whether our response and clarifications have addressed the issues raised in your review, or whether there is anything else we can address.

---

### Official Review · Reviewer_S7vL · 2021-11-01

**Correctness:** 4
**Technical Novelty And Significance:** 4
**Empirical Novelty And Significance:** 4
**Recommendation:** 6
**Confidence:** 4

**Main Review:**

**Strengths**

1. The paper presents a novel, potentially useful setting/objective for the meta-RL/transfer learning communities to focus on, which in some sense bridges objectives pursued by the meta-RL and transfer learning communities.

2. The paper also presents a novel approach to tackle the corresponding problem which would act as a strong baseline for methods tackling the said problem.

3. The paper is also really well written with clear diagrams and algorithm figures which make it very easy to read.

4. I also liked some of the ablations and the choice of baselines as they provide a nice peek at what some of the critical components of the model were that made it click.

5. The paper also does a nice job of tuning the baseline hyperparameters to make sure the results are competitive. I would suggest maybe moving some of those details from Appendix D to the main paper as it might serve as a nice reference for how to perform a fair and competitive evaluation of the baselines!

**Weaknesses**

1. I would like to see more ablations. It was unclear to me from reading the methods section as to why certain design decisions were taken : for example,
    * Why use an on-policy method for online adaptation instead of an off-policy method here as well given you are storing the experiences anyway?
    * Why the specific choice of off-policy method over so many others?
    * What happens when you replace BC with another offline/off policy RL method? Etc.

    In essence I want to understand how much each of the design decisions contribute to the overall performance and which ones were more important than others.

2. I would also like some analysis on why the authors think PPO+TL was able to perform as well as CoMPS (or slightly better) on the meta-world non-stationary tasks. It does seem like a surprising result to me and I think some analysis or discussion there would have been nice! Some hypotheses I have here, for example, are that, maybe the specific non-stationarity ends up providing a kind of curriculum to PPO -TL? Maybe the off-policy updates have too much variance? Maybe the trajectories used for BC update aren't good enough in these types of problems? etc..

3. Related to the point above, I would also like some discussion on the broad class of scenarios where the approach presented in the paper could provide a genuine boost over approaches like PPO+TL, and scenarios where it’s uncertain and why the uncertainty exists (perhaps due to some empirical or numerical issues with the algorithm presented?)

**Other comments**

1. It is also not entirely clear to me if the scenario or motivations presented in the paper for real world applications of the continual RL setting are reasonable? For example, for the home cleaning robot example, it’s hard to imagine that you would ever actually deploy a robot in the real home to collect experiences and learn using RL directly in the sequential setup. You would probably either have some sort of sim2real setup or offline experiences collected in some other (safer?) way used to train the model. I don’t hold this as a major weakness of the work, but I think it would be nice to provide motivations which are more grounded. For example, the disassembling example given at the end of the appendix seems more reasonable to me and might be useful to think along those lines instead!

2. I would also be curious to know how effective the meta-learning objective would be for learning to explore in the new environment when trying to learn the new task (something like “Rothfuss, Jonas, et al. "Promp: Proximal meta-policy search." arXiv preprint arXiv:1810.06784 (2018).”).

3. It might also be a good idea to slightly change how the confidence intervals are computed using suggestions from https://www.google.com/url?q=https://ojs.aaai.org/index.php/AAAI/article/view/11694&sa=D&source=docs&ust=1635372130978000&usg=AOvVaw1BsO05shZBrpwhKwk76dSN (say bootstrap etc)

4. typos/grammatical errors
    * at the same time still adapt to ->  at the same time adapt to
    * rehersal-based methods ->  rehearsal-based methods
    * and the implementation are given ->  and the implementation is given
    * utilize both an improtance-sampled -> utilize both an importance-sampled
    * furthest from previously chosen location -> furthest from the previously chosen location
    * we provide futher analysis -> we provide further analysis
    * behavior of CoMPS is statistical different -> behavior of CoMPS is statistically different
    * due to the increasing difficult of tasks -> due to the increasing difficulty of tasks
    * accelerates acquisition of new tasks -> accelerates the acquisition of new tasks

**Summary Of The Paper:**

The paper presents a new method for continual meta-reinforcement learning called CoMPS. The objective is to quickly achieve a high reward over any sequence of tasks even in non-stationary task distributions. Unlike previous methods in meta-RL, CoMPS uses a hybrid meta-RL approach, where experiences from past tasks are used to learn a fast adaptation procedure (using a meta-RL objective with off-policy inner loop updates and BC outer loop updates). The model is then adapted to any new task using a standard on-policy RL algorithm like PPO. The authors show that this approach outperforms generic meta-learning and transfer learning approaches across various environments for the specific settings considered in the paper.

**Summary Of The Review:**

I like the paper overall, it is clearly written and I think it tries to address an interesting problem. I would however, have liked to see some more ablations and analysis of the approaches and maybe some better motivations too.

---

> ### Author Response · Authors · 2021-11-13
> **Author Response**
>
> Q1: What happens when you replace BC with another offline/off policy RL method? Etc.
>
> A1: It is quite possible that replacing this component with an offline RL component could perform better! Indeed, we hope that our paper will lead to more subsequent work that proposes various improvements and studies how to develop much more effective continual meta-RL methods. We do not claim that our algorithm design is the best one possible, rather we aim to devise a method that can demonstrate that continual meta-RL is actually possible, and that it can accelerate the acquisition of new tasks without revisiting old ones.
>
> Q2: Why the house cleaning example is realistic to motivation for continual meta-RL
>
> A2: We believe that there are many settings in robotics and other domains where revisiting previous tasks is an issue. For example, imagine a robot at a recycling plant that has to disassemble electronics: each item of electronics might be different, and once it has been disassembled, it cannot be disassembled again. So there is a sequence of tasks, prior tasks cannot be revisited, and it is desirable to accelerate all future tasks. Similarly, imagine a house cleaning service performed by a robot that puts away objects -- once the robot has cleaned one house, it goes to a different one. Revisiting houses in a round-robin fashion would be impractical and unnecessary because the houses will already be clean. Even if Sim2Real is used to provide a good starting policy, each house is different and the agent would need to adapt to the changes and house arrangement. This would require a modification to the Sim2Real environment for every new house. In this case, CoMPS would be useful again to improve learning speed in each updated simulation of the real house. While of course, these are not current applications, but rather future applications that require many other components to work well, this seems reasonable in a research paper: not every research paper has to address current applications that are in use, fundamental research can target future applications that are not yet fully feasible.
>
> Q1: Why not use a more off-policy method given that prior data is stored. And “Why the specific choice of off-policy method over so many others?”
>
> A1: This is an important question that gets into the details of why we chose the combination of methods for this research. We investigated a few different methods that can train from prior data. For example, PEARL is a well-known off-policy Meta-RL algorithm. This is why we compare to this algorithm. However, we see that PEARL does not perform well in the continual meta-RL scenario. This is likely due to PEARL’s learned latent model that does not extrapolate well to new tasks. Instead, we want to use a MAML-based approach that can learn to extrapolate if there is similarity between prior tasks and the new task (see, e.g., the discussion of how MAML is expected to degrade more gracefully in the extrapolation regime in [1]). However, MAML is on-policy by design. To overcome this we have improved on a prior MAML-based learning method to now make use of off-policy data and accelerate learning on new tasks. We will update the introduction of the paper to describe the reasoning for these decisions more clear.
>
> Q4: discussion on the broad class of scenarios where the approach presented in the paper could provide a genuine boost over approaches like PPO+TL
>
> A4: Thank you for this question. Meta-learning methods will not be able to generalize to a new task if there is no similarity between the new task and the tasks seen during training. In this case, CoMPS will likely not benefit from meta-training. However, CoMPS still uses RL to solve new tasks and because it is based on MAML it should learn at least as fast as learning the task from scratch from a randomly initialized policy, similar to PPO+TL. This is not the case for prior meta-RL methods that are not MAML-based (like PEARL). The evaluation in the paper illustrates that CoMPS does generalize, and the evidence for this is in figure 6 and 7 that shows CoMPS is able to solve new tasks in fewer iterations than PPO+TL, which uses a similar procedure to train on a new task, but does not benefit from meta-learning (instead just using standard lifelong learning on each task in turn). We will better contextualize the generalization claims in the paper to include these details.
>
> Q5: slightly change how the confidence intervals are computed using suggestions from DeepRL that Matters (say bootstrap etc).
>
> A5: We added details on the bootstrap (30,000) that was used to compute the confidence intervals plotted in the figures. This is also used for the two-sided independent t-test (p-values) evaluation in the appendix (Table 3). Thank you for your ideas on how to improve the analysis.
>
> [1] Finn, Chelsea, and Sergey Levine. “Meta-Learning and Universality: Deep Representations and Gradient Descent Can Approximate Any Learning Algorithm.”  ICLR 2018

---

> > ### Comment · Reviewer_S7vL · 2021-11-21
> > **Response to Author response**
> >
> > I thank the authors for the response. However, I was somewhat disappointed that the authors didn't address my specific suggestions about additional ablations and analysis. Upon further consideration and reading the other discussions I am reducing the score to 6.

---

> > > ### Author Response · Authors · 2021-11-23
> > > **Continued discussion**
> > >
> > > Q1: additional ablations and analysis.
> > >
> > > A1: We apologize for the confusion on this point. We have been processing these experiments. The experiments in the paper can require considerable training time as each method needs to learn 40 different tasks. We have now added this analysis to Appendix H in the updated version of the paper. Here we provide a summary of the results from the analysis. GMPS performs very poorly across all tasks. The addition of PPO (GMPS+PPO) increases performance on \textbf{Cheetah} and \textbf{Metaworld} but the relative performance is still poor. A significant improvement is given with the introduction of off-policy importance sampling (GMPS + PPO + IS). As described in the paper, the importance sampling allows CoMPS to compute unbiased gradient information from a large collection of off-policy data. We also investigate storing additional \textit{skilled data} at the end of learning each new task (GMPS with IS + x2 the skilled data). The addition of more \textit{skilled data} does not generally improve the performance of CoMPS. Following a pattern similar to FTML [1], CoMPS initializes meta-RL training after each task from the policy trained on the previous task, across environments this improves learning speed (CoMPS/ours (GMPS + PPO + IS + FTML)). In particular, the addition of FTML-based training decreases the variation in performance. It results in a smoother reduction in the time used by CoMPS to solve new tasks.
> > >
> > > - [1] Finn, Chelsea, Aravind Rajeswaran, Sham Kakade, and Sergey Levine. 2019. “Online Meta-Learning.” 36th International Conference on Machine Learning, ICML 2019 2019-June: 3398–3410.

---

> > > > ### Comment · Reviewer_S7vL · 2021-11-25
> > > > **Response to Author response**
> > > >
> > > > I appreciate the authors for including some additional ablations and understand that it might be hard to add a bunch of experiments during the rebuttal period given time and compute constraints. I would have bumped my score up to 7 if I had the option, but I would keep it to 6 for now.

---

> > > > > ### Author Response · Authors · 2021-11-26
> > > > > **Thank for the continued discussion**
> > > > >
> > > > > Hello,
> > > > >
> > > > > We are continuing to work on addressing your concerns over the ablation analysis. We have also completed an ablation analysis over the percent of data stored in the off-policy data from each RL task. CoMPS originally stores only 5% of the data generated from RL training on each task. We ablated this value to save 20% and 50% of the total data. We found that the increase in learning data efficiency was minimal and came at a large cost of performance in terms of the time it takes to run these experiments due to the much larger memory footprint. We will include these results in the final version of the paper.
> > > > >
> > > > > We believe this analysis and the comments in our earlier responses address your concerns over the work. If this is not true, we are happy to further discuss the details of CoMPS.

---

> > > > > > ### Comment · Reviewer_S7vL · 2021-11-30
> > > > > > **Final Response**
> > > > > >
> > > > > > I appreciate the authors performing further ablations and the additional experiments on Backward transfer. However, I think the paper could benefit from further ablation analysis. Specifically 2 of the 3 points I raised
> > > > > > 1. Why use an on-policy method for online adaptation instead of an off-policy method here as well given you are storing the experiences anyway?
> > > > > > 3. What happens when you replace BC with another offline/off policy RL method? Etc.
> > > > > >
> > > > > > weren't properly addressed by the ablations (which is fine but I would have liked to at least know why they were not?).
> > > > > > I still think the paper has enough meat to be accepted as a paper in the conference but upon reading the other reviews, I do think the paper needs more careful and well thought out ablations and analysis to be truly valuable.

---

> > > > > > > ### Author Response · Authors · 2021-11-30
> > > > > > > **Responses Appreciated**
> > > > > > >
> > > > > > > Hello,
> > > > > > >
> > > > > > > Thank you for your discussion on the paper and the ablations.
> > > > > > >
> > > > > > > 1. MAML has been shown to have better generalization properties over other methods so it should be best to develop MAML-based algorithms [1]. This is why we base CoMPS on MAML. However, MAML is on-policy by design which makes it challenging to use with off-policy experience. One of the most efficient meta-RL algorithms that fits for off-policy meta-RL training is PEARL. This is why we compare to PEARL and our results show that CoMPS works much better than PEARL in the continual meta-RL setting. We apologize if this was unclear from our earlier comments but we do believe we have provided a fair comparison to a strong off-policy meta-RL method.
> > > > > > >
> > > > > > > 2. It may be possible to investigate additional methods to eventually perform better than CoMPS. However, we have shown that CoMPS currently performs better than other meta-RL algorithms and continual RL algorithms in the problem setting. GMPS appears to be the most sample efficient meta-RL method by using behavior cloning and it is not clear if using off-policy learning will be more sample efficient or stable. However, we are excited about the opportunities for other newer algorithms that can do even better than CoMPS in the future but we believe we have shown that CoMPS is an improvement over prior methods.
> > > > > > >
> > > > > > > - [1] Finn, Chelsea, and Sergey Levine. “Meta-Learning and Universality: Deep Representations and Gradient Descent Can Approximate Any Learning Algorithm.” ICLR 2018

---

### Official Review · Reviewer_FZ5X · 2021-11-02

**Correctness:** 3
**Technical Novelty And Significance:** 1
**Empirical Novelty And Significance:** 3
**Recommendation:** 5
**Confidence:** 3

**Main Review:**

## Merits

1.	This paper studies the problem of continual Meta-RL, which is important and deserves more attention from the community.

2.	The general idea of the proposed CoMPS method is sensible and conceptually simple for addressing the challenges of continual Meta-RL. CoMPS meta-learns on previously seen tasks to acquire a good parameter initialization that fast adapts to new tasks. These adapted policies in turn collect data for meta-learning. The main challenge is that previous tasks cannot be revisited during continual learning, and the policy has to be trained on limited offline data. This challenge is addressed by utilizing importance-sampling policy gradient and off-policy value estimator in the inner loop and behavior cloning in the outer loop. The whole framework is reasonable, and, in some sense, CoMPS can be viewed as an approximate implementation of the FTML algorithm [1].

[1] Finn, Chelsea, et al. "Online meta-learning." International Conference on Machine Learning. PMLR, 2019.

3.	Experimental results look impressive and show that the proposed method achieves superior performance in several sequences of continuous control tasks.

4.	This paper is generally well-written and structured.

## Limitations and Concerns

1.	One major concern is that the technical contribution of the proposed method is very limited. CoMPS is a simple adaptation from the previous method GMPS by using an off-policy variant of PPO in the inner-loop.

2.	Another concern is that CoMPS may not handle well for some badly-learned tasks, as its outer loop objective uses behavior cloning. If previous tasks are not learned well, and the skilled experience dataset $D^*$ contains bad trajectories, the meta-learning procedure may learn an initialization that quickly adapts to imitate the bad trajectories, which harms learning new tasks.

3.	Backward transfer: although the authors claim that they do not evaluate backward transfer or forgetting, it is one of the core challenges in the field of continual learning. It is interesting to see how CoMPS performs on previous tasks, which may contribute to a deeper understanding of CoMPS.

4.	Related works: most related works have been adequately discussed. However, several works incorporating meta-learning into continual supervised learning are missing [2,3]. A discussion over these works should be added.
[2] Joseph, K. J., and Vineeth N. Balasubramanian. "Meta-consolidation for continual learning." arXiv preprint arXiv:2010.00352 (2020).
[3] Gupta, Gunshi, Karmesh Yadav, and Liam Paull. "La-maml: Look-ahead meta-learning for continual learning." arXiv preprint arXiv:2007.13904 (2020).

5.	Experimental details: environment descriptions of Ant Direction and MetaWorld in the caption of Figure 5 do not seem to correspond to Appendix A, which is confusing.

6.	Other minor things: at the start of each meta-learning procedure (Algorithm 1), how is $\theta$ initialized? Do you use the $\theta$ achieved at the end of the last meta-learning procedure, or do you use the parameters updated after the RL procedure?


**Summary Of The Paper:**

This paper proposes a method that incorporates fast adaptation into continual learning. The main challenge is that previous tasks cannot be revisited during continual learning, and the policy has to be trained on limited offline data. The authors address this challenge by utilizing importance-sampling policy gradient and value estimators in the inner loop and behavior cloning in the outer loop, which enables meta-training with fully offline data. Experimental results show that their method achieves superior forward transfer compared to prior methods.

**Summary Of The Review:**

Overall, this paper is an interesting step towards solving the problem of continual Meta-RL. Its presentation is clear, and the proposed general framework is reasonable. One major concern is its novelty, which resembles the previous approach GMPS. Although several key limitations have been identified in the discussions of this paper, addressing some of them is important and can significantly strengthen this paper, particularly requiring the data storage of previous tasks and no mechanism for handling forgetting.

---

> ### Author Response · Authors · 2021-11-13
> **Author Response**
>
> Our main contribution is an algorithm to address the continual meta-reinforcement learning problem. This problem setting, which has not to our knowledge been defined or studied before, does not allow for revisiting previously seen tasks, and requires the agent to acquire each new task as quickly as possible. We argue that this setting is realistic in practice, since RL agents typically would not be able to revisit previously experienced environments easily. In terms of the actual algorithm, CoMPS makes three changes to GMPS: (1) using a PPO-style inner loop policy gradient update; (2) using V-trace to enable off-policy learning with old data from prior tasks, which cannot be revisited and (3) an FTML type training process where CoMPS trains a meta-policy to initialize RL task learning that is used to initialize the following meta-training. The experiment results in the paper show the importance of these components. That said, our main contribution is *not* the particular set of components (though these components are important), but rather the formulation of the problem and a novel empirical demonstration that continual meta-RL in this setting accelerates acquisition of each new task. We believe this is significant, and could spur future research on continual meta-reinforcement learning methods.
>
> Q0: Although several key limitations have been identified in the discussions of this paper, addressing some of them is important and can significantly strengthen this paper, particularly requiring the data storage of previous tasks and no mechanism for handling forgetting.
>
> A0: We agree that dealing with forgetting (stability) is an important component of the continual learning problem. However, CoMPS is designed to address the challenge of learning new tasks quickly (plasticity). CoMPS must still retain prior data, and we do not claim to fix this problem, though it may be important to tackle in future work. We have however shown across multiple environments that CoMPS provides an improvement over prior methods in terms of plasticity, and we believe this is significant. We appreciate the reviewers’ concerns and we will make it clear much earlier in the paper that our work focuses on forward transfer, that retaining prior data is one of its limitations, and that we leave a full treatment of the forgetting problem for future work. However, we believe this scoping is reasonable, since in many settings it is possible to retain prior data, as is done in several rehearsal-based prior works on supervised online learning [2, 3, 4, 5]. In addition, considering the availability of cheap disk space the need to consider the problem of forgetting is lessening. This makes it odd to focus on this non-issue when the community can instead focus on the problem of how to learn new tasks faster (plasticity).
>
> Q1: CoMPS is an approximation of FTML
>
> A1: FTML focuses on supervised learning as an online learning problem and CoMPS is instead focused on reinforcement learning tasks. It is true that CoMPS can be seen as an approximate FTML that includes many necessary changes for solving RL tasks. We have added content to the beginning of section 4 to note this connection.
>
> Q2: CoMPS may perform poorly if the data collected on prior tasks was not of sufficient quality.
>
> A2: This question has two important parts. First, CoMPS uses success thresholds as a means of forcing the agent to sufficiently solve tasks before moving on to new tasks. This helps increase the quality of the collected data. The maximum amount of time given for each task can be increased to ensure the agent has enough time to solve the task. Second, even if the prior task was not solved the data from that task is likely to be unique and add diversity to the meta training dataset which will help with learning more generalizable parameters [1]. That said, we acknowledge that this may be a limitation in some settings, and we’ve added discussion of this to the paper along with the discussion of success thresholds.
>
> Q3: What is the backwards transfer performance of CoMPS.
>
> A3: CoMPS does not claim to provide a solution to the backwards transfer problem (forgetting) and instead focuses on the forward transfer problem (plasticity). The results in the paper show that CoMPS performs better in this respect to prior meta-learning and continual learning methods. However, we will provide a graph of the backwards performance during meta-training which we agree will help us understand the strengths and future directions for CoMPS.
>
> Q4: Prior work
>
> A4: Thank you for the noted prior work that uses meta-learning in a continual learning setting. However, these suggested prior works have not been used for RL problems. They would require updates to incorporate RL and off-policy training for efficiency that would likely result in an algorithm very similar to CoMPS. We will include these references in the paper.

---

> ### Author Response · Authors · 2021-11-13
> **Author Response Continued**
>
>
>
> Q5: At the start of each meta-learning procedure (Algorithm 1), how is initialized?
>
> A6: We use a method very similar to FTML. The meta-training is initialized from the most recent RL policy parameters learned on the prior task. We will update the algorithm to make this more clear.
>
> - [1] Gupta A, Eysenbach B, Finn C, Levine S. Unsupervised meta-learning for reinforcement learning. arXiv preprint arXiv:1806.04640. 2018 Jun 12.
> - [2] Riemer, Matthew, Ignacio Cases, Robert Ajemian, Miao Liu, Irina Rish, Yuhai Tu, and Gerald Tesauro. 2018. “Learning to Learn without Forgetting by Maximizing Transfer and Minimizing Interference.” arXiv [cs.LG]. arXiv. https://doi.org/arXiv:1810.11910v1.
> - [3] Rolnick, David, Arun Ahuja, Jonathan Schwarz, Timothy P. Lillicrap, and Greg Wayne. 2018. “Experience Replay for Continual Learning.” arXiv [cs.LG]. arXiv. http://arxiv.org/abs/1811.11682.
> - [4] Atkinson, Craig, Brendan McCane, Lech Szymanski, and Anthony Robins. 2021. “Pseudo-Rehearsal: Achieving Deep Reinforcement Learning without Catastrophic Forgetting.” Neurocomputing 428 (March): 291–307.
> - [5] Isele, David, and Akansel Cosgun. 2018. “Selective Experience Replay for Lifelong Learning.” Proceedings of the AAAI Conference on Artificial Intelligence 32 (1). https://ojs.aaai.org/index.php/AAAI/article/view/11595.

---

> ### Author Response · Authors · 2021-11-18
> **Author response**
>
> Q3.1: Backward transfer analysis for CoMPS
>
> A3.1: We have performed an analysis of the backwards transfer of CoMPS. For the complete analysis please see Section G in the appendix. We remind the reviewers that we believe the contributions of CoMPS in terms of forward transfer as mentioned in our response is sufficient for acceptance. Here we provide a summary of the findings from the analysis. We can see from these plots that \methodName exhibits strong backwards transfer on the \textbf{Ant Direction}, \textbf{Ant Goal}, and \textbf{Cheetah} environments. For \textbf{Ant Goal} and \textit{Ant Direction} backwards transfer appears to stabilize, indicating that learning new tasks may not affect performance on old tasks. This observation aligns with the properties of meta-learning and MAML such that adding more tasks generally increases adaptation performance. The reducing performance on \textbf{Cheetah} is likely related to the increasing difficulty of tasks. CoMPS demonstrates backwards transfer on \textbf{Metaworld} initially, however, this reduces as more tasks are seen, again indicating the difficulty of discovering generalizable structure across the diverse \textbf{Metaworld} tasks.

---

> ### Author Response · Authors · 2021-11-25
> **Follow up discussion**
>
> Dear Reviewer, We hope that you've had a chance to read our response. We would really appreciate a reply as to whether our response and clarifications have addressed the issues raised in your review, or whether there is anything else we can address.

---

> ### Comment · Reviewer_FZ5X · 2021-11-29
> **Feedback on the response**
>
> I appreciate the author's clarification and their efforts on this work. I think this work is an interesting step towards continual Meta-RL, but I am still concerned about its technical novelty. Addressing several key limitations discussed in my comments and in the submission itself will improve this paper. For now, I will keep my rating.

---

### Official Review · Reviewer_8AHL · 2021-11-03

**Correctness:** 3
**Technical Novelty And Significance:** 2
**Empirical Novelty And Significance:** 3
**Recommendation:** 6
**Confidence:** 3

**Main Review:**

My main observation is that the paper reveals possible novelty and innovations very late and unclearly. The abstract explains the problem that the algorithm attempts to solve, but reveals no insight about how this happens, and thus does not provide any concrete claim. This issue continues in the introduction where the contribution part focuses on what problem is being solved, but not how. What are the main scientific and technological innovation that are introduced with respect to existing approaches? Eventually, section 4 (CoMPS overview) enters in the description of the proposed algorithm, but even the following sections 4.1 and 4.2 do not clearly state what specific novel aspects are introduced with respect to existing methods, and which are re-used or adapted, which makes it very difficult to appreciate the novel contributions and originality of the work. In section 4.3, it is mentioned that the proposed method uses "both an improtance-sampled policy gradient estimator and an importance-sampled value estimate for the baseline in the policy gradients” and it claims that the following ablation study demonstrate it’s importance. Is this one of the major contribution of the paper?

Similarly, the idea that the novel algorithm uses an alternation of off-policy (for the inner loop) and imitation learning (for the outer loop) transpires without sufficient clarity and related claims. An interesting hypothesis, e.g. is mentioned in section 4.2 : " In contrast to methods that are concerned with forgetting, the parameters produced by this meta-RL training can quickly learn new behaviors that are similar to the high-value policies from previous tasks and, if enough prior tasks have been seen, likely generalize to quickly learn new tasks as well. However, this is not sufficiently followed up in the experimental analysis. How does this mechanism affect performance in the various tasks? What can we learn from the fact that CoMPS performs better than the baselines on the first three tasks, but it's comparable to PPO+TL in the arguably more challenging MetaWorld tasks?


**Summary Of The Paper:**

This paper proposes a meta-RL algorithm that is focused in particular on the learning of sequential tasks without revisiting previously seen tasks. This approach, as stated in the paper, could be particularly beneficial in real-world applications and robotics where it might be unfeasible to revisit previously seen tasks. The paper builds on a successful meta-RL algorithm (MAML) and appears to leverage on the use of off-policy methods to replay previous data from previous policy without re-engaging with older tasks.


**Summary Of The Review:**

In summary, the paper appears to have solid basis in the literature and tackles an important problem in meta-RL, and it appears to propose an interesting combination of known approaches to solve the stated problem of sequential-task meta-RL without task repetition. The experimental results are encouraging. However, it fails to make clear claims or clearly explain the key novel aspects of the algorithm, how and why they are sufficiently novel.

---

> ### Author Response · Authors · 2021-11-13
> **Author Response**
>
> The primary contribution of our work is to demonstrate that a continual meta-RL method that we propose can accelerate the acquisition of new tasks in a setting where prior tasks cannot be revisited. The primary contribution is not this specific algorithm (future work may propose new continual meta-RL methods), but rather the novel empirical demonstration that continual meta-RL accelerates task acquisition, and the introduction of the continual meta-RL problem statement itself. However, the specific algorithm we propose makes three changes to GMPS that result in important performance improvements: (1) using a PPO-style inner loop policy gradient update; (2) using V-trace to enable off-policy learning with old data from prior tasks, which cannot be revisited and (3) an FTML[1] type training process where CoMPS trains a meta-policy to initialize RL task learning that is used to initialize the following meta-training. We believe these contributions are significant, and successful ICLR submissions do not necessarily need to focus on complex new technical components, but could also consist of novel empirical results (“empirical novelty and significance”), which we believe our paper satisfies by demonstrating for the first time that continual meta-RL can accelerate acquisition of new tasks without revisiting old ones.
>
> Q1: if enough prior tasks have been seen, likely generalize to quickly learn new tasks as well. However, this is not sufficiently followed up in the experimental analysis
>
> A1: Thank you for this question. Meta-learning methods will not be able to generalize to a new task if there is no similarity between the new task and the tasks seen during training. In this case, CoMPS will likely not benefit from meta-training. However, CoMPS still uses RL to solve new tasks and because it is based on MAML it should learn at least as fast as learning the task from scratch from a randomly initialized policy, similar to PPO+TL. This is not the case for prior meta-RL methods that are not MAML-based (like PEARL). The evaluation in the paper illustrates that CoMPS does generalize, and the evidence for this is in Figure 6 and 7 that shows CoMPS is able to solve new tasks in fewer iterations than PPO+TL (and other prior methods), which uses a similar procedure to train on a new task, but does not benefit from meta-learning (instead just using standard lifelong learning on each task in turn). We will better contextualize the generalization claims in the paper to include these details.
>
> Q2: “What can we learn from the fact that CoMPS performs better than the baselines on the first three tasks, but it's comparable to PPO+TL in the arguably more challenging MetaWorld tasks”
>
> A2: This is an important and nuanced question. However, in general, CoMPS is shown to perform the best on most of the benchmark environments in the paper. The reduced performance on Metaworld is a sign of the difficulty of the Metaworld environments, which was also noted in the Metaworld paper. However, while CoMPS and other methods may perform similarly in terms of reward achieved on non-stationary task sequences for Metaworld Figure 11 shows that CoMPS is able to achieve these rewards in fewer iterations. This is particularly clear for CoMPS vs GMPS+PPO, showing the improvement from the addition of off-policy importance sampling. We will include this discussion and connection in the paper. Overall however we believe that this shows that the continual meta-RL problem is far from solved, and more research on this topic and meta-RL, in general, is needed. We hope that our work will encourage the community to tackle these challenges, and we hope that the reviewers appreciate our attempt to provide a forthright and realistic assessment of our method’s performance.
>
> [1] Finn, Chelsea, Aravind Rajeswaran, Sham Kakade, and Sergey Levine. 2019. “Online Meta-Learning.” 36th International Conference on Machine Learning, ICML 2019 2019-June: 3398–3410.

---

> > ### Comment · Reviewer_8AHL · 2021-11-18
> > **Response**
> >
> > I thank the authors for providing additional comments and explanations. I agree that a valid submission does not have to include completely new methods, and it can also focus on empirical studies and verification of existing approaches and/or a combination of those. But this has to be made very clear so that the scope and the claims of the paper are illustrated from the start. The present method is indeed such a combination of existing approaches: the authors are working in the right direction and I am willing to update the assessment. I believe this particular research area will see considerable advancements in the following years: for this paper to achieve a higher score, it should include more in-depth experiments and analysis and a clearer narrative from the start. The newer revisions of the paper have very minor changes, while I believe the paper requires considerable revision.

---

> > > ### Author Response · Authors · 2021-11-19
> > > **Discussion**
> > >
> > > Re: Scope
> > >
> > > We are working on revising the paper to make the claims around the problem setting more clear.
> > >
> > > Re: more in-depth experiments and analysis
> > >
> > > We already conduct comparisons with $5$ distinct prior methods and baselines on four distinct task sequences, and with two different task distributions. We think that this analysis provides a thorough comparison in this setting, especially accounting for the fact that there are (to our knowledge) no prior methods that address this problem setting. However, we would be happy to add additional analysis that the reviewer would consider appropriate -- here, we would really appreciate any suggestions that the reviewer would have about what kind of analysis would be needed here.

---

> > > > ### Author Response · Authors · 2021-11-25
> > > > **Follow up discussion**
> > > >
> > > > Dear Reviewer, We hope that you've had a chance to read our response. We would really appreciate a reply as to whether our response and clarifications have addressed the issues raised in your review, or whether there is anything else we can address.

---

> > > > > ### Comment · Reviewer_8AHL · 2021-11-29
> > > > > **Response**
> > > > >
> > > > > I believe the authors have put considerable effort to improve the paper according to this reviewer's feedback. In particular, the scope and claims are now clearer. I have already increased the evaluation by one, and I'm happy to increase it again.

---

> > > > ### Author Response · Authors · 2021-11-26
> > > > **Continued discussion**
> > > >
> > > > Hello and thank you for your continued discussion. We have edited the paper to make the claims of the paper more clear. We note, the original paper did not include a specific statement that focused on the proposal of the continual meta-RL problem statement as the primary contribution. Instead, the paper states "Our primary contribution is a meta-reinforcement learning algorithm that supports the sequential multi-task learning setting, where the agent cannot revisit previous tasks to collect data." Which we believe fits our contribution well. Now we have also added extensive analysis and discussion of the CoMPS algorithm in terms of backwards transfer (Appendix G) and ablation analysis (Appendix H). We believe this clears up the issues around the details, experiments and claims in the paper. If this is not true we are happy to discuss further details.

---

> ### Author Response · Authors · 2021-11-18
> **Feedback**
>
> Hello reviewer 8AHL, we would be grateful if you can confirm whether our responses have addressed your concerns, and let us know if any issues remain. To recap our response, we:
>
> - Discussed the contributions of CoMPS over prior work
> - Provided details on claims in the paper
> - Included more discussion on the performance of CoMPS.

---

### Public Comment · ~Massimo_Caccia1 · 2021-11-09
**code release?**

Hi,
Interesting work!
Any chance we can get our hands on the code?
Thanks

---

### Decision · Program_Chairs · 2022-01-20

**Decision:**

Accept (Poster)

**Comment:**

This paper develops a novel continual meta-reinforcement learning algorithm that focuses on learning sequential tasks without revisiting previous tasks. The setting is compelling, and the method is well-developed with good empirical results. The initial version of the paper included a variety of issues, especially lack of clarity in some aspects and the contributions, that were remedied through discussion with the reviewers and subsequent revisions. The discussion among the reviewers seems to have settled on leaning toward a weak accept overall, with one low score that should be dismissed claiming lack of novelty (which isn't correct - the paper certainly is sufficiently novel).  There do remain some concerns by two reviewers that although "the paper has enough meat to be accepted, ... [it] needs more careful and well thought out ablations and analysis to be truly valuable." Although the authors have revised the paper to address this issue of a precise analysis, adding material into the appendices with some changes to the main text, they are encouraged to make certain that these aspects are integrated and clear throughout.